# Theta oscillations represent collective dynamics of multineuronal membrane potentials of murine hippocampal pyramidal cells

Asako Noguchi [1✉], Kotaro Yamashiro[1], Nobuyoshi Matsumoto [1,2] & Yuji Ikegaya [1,2,3]

Theta (θ) oscillations are one of the characteristic local field potentials (LFPs) in the hippocampus that emerge during spatial navigation, exploratory sniffing, and rapid eye movement sleep. LFPs are thought to summarize multineuronal events, including synaptic currents and action potentials. However, no in vivo study to date has directly interrelated θ oscillations with the membrane potentials ($V$m) of multiple neurons, and it remains unclear whether LFPs can be predicted from multineuronal $V$ms. Here, we simultaneously patch-clamp up to three CA1 pyramidal neurons in awake or anesthetized mice and find that the temporal evolution of the power and frequency of θ oscillations in $V$ms ($\theta_{Vm}$s) are weakly but significantly correlate with LFP θ oscillations ($\theta_{LFP}$) such that a deep neural network could predict the $\theta_{LFP}$ waveforms based on the $\theta_{Vm}$ traces of three neurons. Therefore, individual neurons are loosely interdependent to ensure freedom of activity, but they partially share information to collectively produce $\theta_{LFP}$.

[1] Graduate School of Pharmaceutical Sciences, The University of Tokyo, Tokyo 113-0033, Japan. [2] Institute for AI and Beyond, The University of Tokyo, Tokyo 113-0033, Japan. [3] Center for Information and Neural Networks, National Institute of Information and Communications Technology, Suita City, Osaka 565-0871, Japan. ✉email: asakonoguchi.an@gmail.com

ocal field potentials (LFPs) are associated with various aspects of animal cognition and behavior, including attention, volition, and learning[1–4], and are also used as biomarkers of pathological states[5,6]. One of the major components of LFPs in the hippocampi of humans[7–9] and rodents[10–12] is theta (θ) oscillations (3–10 Hz). θ oscillations occur mainly during active exploration, exploratory sniffing, and rapid eye movement sleep and are essential for the normal functioning of the hippocampus[12–14]. Indeed, θ oscillations are correlated with memory performance[15–19], and reduced θ oscillations are associated with memory deficits[19,20], whereas enhanced θ oscillations facilitate cognitive function[21,22].

As a component of LFPs, θ oscillations are thought to reflect extracellular currents that arise from the collective dynamics of subthreshold membrane potentials ($Vm$)[23–25], including synaptic activity, dendritic integration, action potentials, afterpotentials, and other channel-dependent neuronal events, in a large number of neurons[26,27]. However, it remains unclear how θ oscillations are associated with the $Vm$ dynamics of multiple neurons in vivo. Previous studies have examined the relationship between LFPs and the $Vm$ of a single pyramidal cell and reported both similarities and discrepancies[23,28,29]. For example, one study with simultaneous LFP and single intracellular recordings from the hippocampi of anesthetized rabbits reported that intracellular θ oscillations ($θ_{Vm}$) occurred together with LFP θ oscillations ($θ_{LFP}$) at similar oscillation frequencies[23], suggesting that $θ_{Vm}$ serve as a source of $θ_{LFP}$; however, not all neurons exhibited $θ_{Vm}$ during $θ_{LFP}$ states, and the amplitudes of $θ_{Vm}$ were also heterogeneous across cells. Thus, it remains unclear how similar $θ_{Vm}$ rhythms are exhibited in different cells during a $θ_{LFP}$ period. A more recent study examined the relationships between $θ_{LFP}$ and $θ_{Vm}$ in head-fixed running mice[28], demonstrating that although $θ_{Vm}$ tended to occur together with $θ_{LFP}$, these two types of oscillations often had inconsistent phases and frequencies. This inconsistency is functionally important for θ phase precession because a $θ_{Vm}$ rhythm that is faster than the $θ_{LFP}$ rhythm allows the neuron to fire at earlier phases of $θ_{LFP}$ during spatial movement. Nonetheless, it remains unclear how the diverse $θ_{Vm}$s of multiple hippocampal neurons are reflected in unidimensional $θ_{LFP}$. These questions can be answered, at least in part, by simultaneously recording the $Vm$s of multiple neurons.

In the present study, we simultaneously recorded the $Vm$s of up to three CA1 pyramidal cells in mice to directly compare their $Vm$ dynamics to LFPs. We found that $θ_{Vm}$ occurred intermittently, often together with $θ_{LFP}$. In some but not all LFP-cell pairs and cell-cell pairs, θ oscillations showed similar changes in power and frequency over time. Specifically, the $θ_{LFP}$ power increased when more CA1 pyramidal neurons simultaneously emitted $θ_{Vm}$s. When neurons had similar $θ_{Vm}$ frequencies, the $θ_{LFP}$ power increased, and the $θ_{LFP}$ frequency approached the average value of the neuron $θ_{Vm}$ frequencies. As a result, $θ_{LFP}$ was predictable from the $θ_{Vm}$ dynamics of three CA1 pyramidal cells using deep learning. These findings provide fundamental insights into how LFPs are associated with multineuronal $Vm$s in vivo, demonstrating their collective potential for predicting LFPs.

## Results

### Coordinated $θ_{LFP}$s across the dorsal hippocampal CA1 area. 
To ensure the $θ_{LFP}$-$θ_{Vm}$ correlation analyses below, we first verified whether $θ_{LFP}$s were uniform within our targeted area. LFPs were simultaneously recorded from four sites in the dorsal CA1 area of the mouse hippocampus (Fig. 1a–c). The area enclosed by the four sites encompassed the area for LFP and patch-clamp recordings in the following experiments. LFPs spontaneously alternated

between periods with and without $θ_{LFP}$ at frequencies ranging from 3 to 10 Hz (Fig. 1d, e). The $θ_{LFP}$ power was reduced after intraperitoneal administration of 50 mg/kg atropine, a muscarinic receptor antagonist (Supplementary Fig. 1). Therefore, the $θ_{LFP}$ represented type 2 θ oscillations[30].

The LFP traces were divided into a time series of 1-s segments, and the mean $θ_{LFP}$ power was calculated for each segment. The time-dependent changes in the mean $θ_{LFP}$ power were plotted in the space of each pair of simultaneously recorded LFP traces (Fig. 1f). The more similar the changes in the $θ_{LFP}$ power of the two recording sites were, the closer the data points were to the identity line in the plot. To quantify this similarity, we calculated the correlation coefficient for each plot. For all 30 LFP pairs from 5 mice, the correlation coefficients were significant and positive (Fig. 1g), indicating that $θ_{LFP}$ emerged simultaneously at all four recording sites. Considering that the recording time affects the correlation, the entire recording period was divided into 1-, 2-, 3-, 5-, 10- or 15-min time windows, and the correlation was evaluated for each time window. While the correlation increased slightly with increasing time window lengths, a significant positive correlation was observed for >90% of the entire recording period, regardless of the length of the time window (Supplementary Fig. 2, all time window lengths, $P = 1.1 \times 10^{-4}$, $\chi^2 = 25.6$; time windows >2 min, $P = 0.049$, $\chi^2 = 9.6$; time windows >3 min, $P = 0.11$, $\chi^2 = 6.11$, chi-square test, $n = 900$, 450, 300, 180, 90, and 60 time windows for 1, 2, 3, 5, 10, and 15 min, respectively). Thus, the time changes in $θ_{LFP}$ power were synchronized among the four recording sites and were minimally affected by the recording time. To examine the phase relationship between simultaneously recorded $θ_{LFP}$s, the LFP traces were bandpass filtered between 3 and 10 Hz, and the cross-correlations were computed. The time lags were calculated by referencing the $θ_{LFP}$s recorded from relatively medial or anterior locations for individual pairs so that the $θ_{LFP}$ propagations along the longitudinal axis of the hippocampus[11,31] could be extracted (Fig. 1h, i). Overall, the $θ_{LFP}$s were highly synchronous between LFP pairs, regardless of their relative positions (Fig. 1h, i). The $θ_{LFP}$s recorded from medial positions significantly preceded their counterparts (Fig. 1j ML, $P = 0.0023$ vs. time 0, $t_{14} = -3.7$, Student's t-test, $n = 15$ pairs from 5 mice), while no such effects were observed between $θ_{LFP}$ pairs along the anteroposterior axis (Fig. 1j AP, $P = 0.095$ vs. time 0, $t_{13} = 1.8$, Student's t-test, $n = 14$ pairs from 5 mice), which is consistent with a previous report[11]. Therefore, $θ_{LFP}$s were synchronized across the entire hippocampal windows targeted in our study, with mediolateral propagation on a fine time scale.

Next, we divided the entire recording times into $θ_{LFP}$ and non-$θ_{LFP}$ periods using a threshold based on the standard deviation (SD) of the background noise against the power of the θ frequencies in the wavelet spectrogram (Supplementary Fig. 3). To determine an appropriate SD threshold, we categorized putative $θ_{LFP}$ and non-$θ_{LFP}$ periods using various multiples of the SD (1 SD, 2 SDs, 3 SDs, and 4 SDs) and calculated the Dice similarity coefficients to estimate the strength of synchronization of the detected $θ_{LFP}$ periods between two recording sites (Supplementary Fig. 3a). The distributions of the Dice similarity coefficients were compared to their statistical chance levels obtained from 10,000 surrogate data samples in which the detected $θ_{LFP}$ periods were randomly shuffled across time within each recording site. The difference in the original and surrogate data distributions was assessed according to the D value of a two-sample Kolmogorov–Smirnov test, and the D value reached the maximum at 2 SDs (Supplementary Fig. 3b). Therefore, in the following analyses, we defined a θ period as a time period during which the θ power continued to exceed 2 SDs against the background noise.

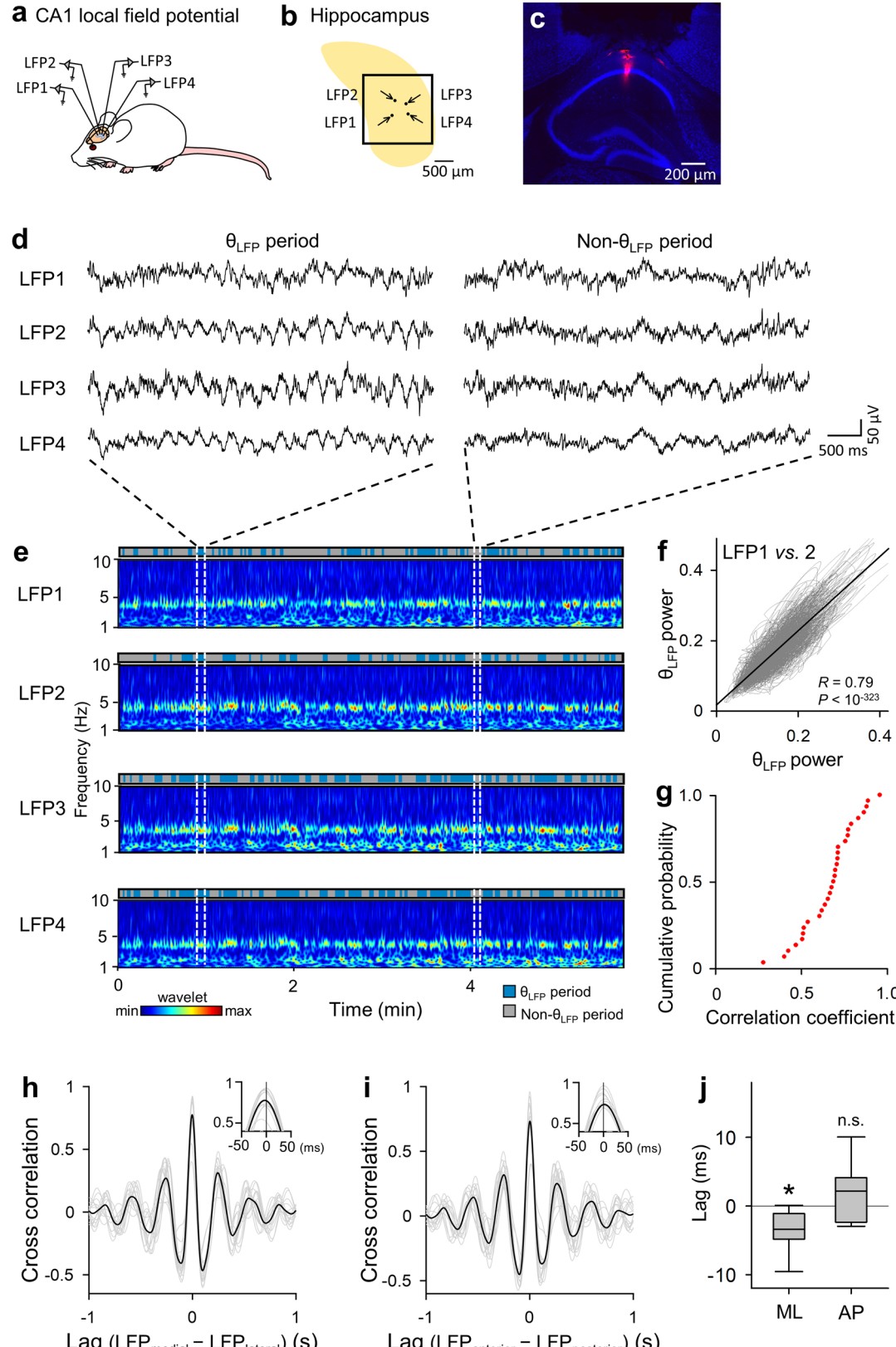

**Characteristics of Vm activity during θ$_{Vm}$ and non-θ$_{Vm}$ periods**. To examine the θ$_{LFP}$-θ$_{Vm}$ relationship, we first characterized the θ$_{Vm}$ dynamics. We patch-clamped pyramidal cells in the hippocampal CA1 region of urethane-anesthetized mice (Fig. 2a). Recordings were excluded from subsequent analyses if *post hoc* biocytin-based visualizations, their recording sites, and/or their

firing properties failed to identify the recorded neurons as CA1 pyramidal cells (Fig. 2b). As a result, we recorded Vms from 220 patch-clamped cells in a total of 112 mice. The recording periods ranged from 30 s to 2097 s (median = 180 s).

Similar to LFPs, spontaneous Vm responses alternated between periods with and without θ oscillations (Fig. 2c); of the total

**Fig. 1 Synchronized $\theta_{LFP}$s across the dorsal hippocampal CA1 area. a** Schematic illustration of simultaneous in vivo LFP recordings from four sites in the dorsal hippocampal CA1 area. **b** Representative top view schematic of the hippocampus (yellow), hippocampal window (square), recording sites (black dots), and directions of the inserted glass pipettes used to acquire the data shown in **d**. **c** Fluorescence image of the track of an LFP electrode visualized by DiI (red). The histological section was counterstained with fluorescent Nissl stain (blue). **d** Representative traces of LFPs recorded simultaneously from four sites during a $\theta_{LFP}$ period (left) and a non-$\theta_{LFP}$ period (right). **e** Wavelet spectrograms of four simultaneously recorded LFP traces, parts of which are shown in **d**. Blue and gray bars indicate $\theta_{LFP}$ and non-$\theta_{LFP}$ periods, respectively. **f** Temporal evolution of the relationship between the $\theta_{LFP}$ powers of LFP1 and LFP2 shown in **e**. For a given 1-s segment, the $\theta_{LFP}$ powers of two LFPs are plotted as a single dot in the space of LFP1 and LFP2; temporally adjacent dots are connected by gray lines. The significant positive correlation indicates that the two $\theta_{LFP}$ powers changed similarly over time ($R = 0.79$, $P < 10^{-323}$, $t_{17,992} = 172.8$, $t$-test for correlation coefficients, $n = 17,994$ segments). The black lines indicate the lines of best fit with ordinary least-squares regression. **g** Cumulative probability distribution of the correlation coefficients between pairs of $\theta_{LFP}$ powers, as calculated in **f**. The correlations were statistically significant for all 30 pairs of 20 LFP traces recorded from 5 mice. **h** Cross-correlograms of the 14 pairs of simultaneously recorded LFP traces bandpass filtered at 3–10 Hz (gray line) and their mean (black). Only the pairs of LFPs for which the relative positions of the electrodes along the mediolateral axis could be identified were included. In the inset, the time scale is expanded near 0 ms, and the plot indicates that the $\theta_{LFP}$s recorded from relatively medial recording sites preceded their counterparts. **i** Same as **h**, but for the 15 pairs of LFPs for which the relative positions of the electrodes along the anteroposterior axis could be identified. Cross-correlations peaked at 0-ms time lags. **j** Time lags between pairs of $\theta_{LFP}$s were calculated for the mediolateral (ML) and anteroposterior (AP) pairs shown in **h** and **i**, respectively. $\theta_{LFP}$ propagation was observed in the medial to lateral direction ($P = 0.0023$ vs. time 0, $t_{14} = -3.7$, Student's $t$-test, $n = 15$ pairs from 5 mice) but not along the anteroposterior axis ($P = 0.095$ vs. time 0, $t_{13} = 1.8$, Student's $t$-test, $n = 14$ pairs from 5 mice).

recording time, ~$34 \pm 26\%$ represented $\theta$ periods (mean $\pm$ SD of 220 cells, ranging from 0.84 to 95%). The frequency of $\theta_{Vm}$ was not unique to a given cell but varied among $\theta_{Vm}$ periods (Fig. 2d). Spikes from pyramidal cells were entrained to $\theta_{Vm}$ cycles such that the firing rates peaked at the $\theta_{Vm}$ phase of 0° (Fig. 2e; $P < 10^{-323}$, $Z = 3.7 \times 10^3$, Rayleigh test, $n = 4655$ spikes from 220 cells). Analysis using a generalized linear mixed model demonstrated that the mean firing rates during $\theta_{Vm}$ periods increased as a function of $\theta_{Vm}$ power (Fig. 2f, $\beta = 0.22$, $P = 8.0 \times 10^{-9}$, $t_{1032} = 5.8$, $n = 1,034$ $\theta$ periods from 220 cells). The mean firing rates also increased with increasing $\theta_{Vm}$ frequency (Fig. 2g, $\beta = 0.10$, $P = 0.035$, $t_{1,032} = 2.1$, $n = 1034$ $\theta$ periods from 220 cells). The firing rates and power were $Z$-standardized in each cell and pooled across all recorded cells.

**Weakly correlated $\theta_{LFP}$ and $\theta_{Vm}$.** We next compared $\theta$ oscillations between $Vm$ and LFPs. While recording $Vm$ from a single CA1 pyramidal cell, we recorded LFPs from a single site in the CA1 region in the same hippocampal window (Fig. 3a–e). The $\theta_{LFP}$ power was not correlated with the firing rates of patch-clamped cells (Supplementary Fig. 4). To examine whether $\theta_{Vm}$ and $\theta_{LFP}$ occurred simultaneously, we plotted the time-dependent changes in the $\theta$ powers of $Vm$ and LFPs every 100 ms (Fig. 3c, e). The correlation coefficients varied among cells; the $\theta_{Vm}$ power exhibited a significant positive correlation with the $\theta_{LFP}$ power in 66 (41%) cells out of 160 neurons, whereas the remaining 94 (59%) cells did not exhibit significant correlations (Fig. 3f). Therefore, the $\theta_{Vm}$ power was correlated, at least in part, with the $\theta_{LFP}$ power; however, this correlation was not robust. The correlation coefficients were not associated with the distances between the LFP recording sites and the patch-clamped cells (Fig. 3g, $R = 0.087$, $P = 0.43$, $t$-test for correlation coefficients, $n =$ all 86 cells whose loci were confirmed *post hoc*), indicating that even if the cells were located near the LFP recording site or generated strong $\theta_{Vm}$, their influence on $\theta_{LFP}$ was not necessarily large. In addition, there were no significant relationships between the correlation coefficients and the recording time length (Supplementary Fig. 5a, $R = 0.11$, $P = 0.19$, $t$-test for correlation coefficients, $n = 160$ cells), the mean $\theta_{Vm}$ power (Supplementary Fig. 5b, $R = -0.042$, $P = 0.60$, $n = 160$ cells), the mean firing rate (Supplementary Fig. 5c, $R = -0.081$, $P = 0.31$, $n = 160$ cells), the mean $Vm$ (Supplementary Fig. 5d, $R = -0.14$, $P = 0.076$, $n = 160$ cells), and the standard deviation (SD) of $Vm$ (Supplementary Fig. 5e, $R = 0.068$, $P = 0.39$, $n = 160$ cells). Furthermore, neither the cell locations along the proximodistal (Supplementary Fig. 5f,

$P = 0.62$, one-way analysis of variance (ANOVA), $n = 27, 41, 45$, 11, 11 for CA1a, a/b, b, b/c, c, respectively) nor the radial (Supplementary Fig. 5g, $P = 0.27$, $t_{131} = 1.1$, Student's $t$-test, $n = 80$ and 53 cells for deep and superficial, respectively) axes were significantly related to the correlation coefficients. Therefore, the engagement of each cell with the ongoing $\theta_{LFP}$ might not be intrinsically predetermined but rather be flexibly modifiable.

During a co-$\theta$ period when $\theta$ oscillations occurred simultaneously in $Vm$ and LFPs, the $\theta_{Vm}$ frequency was positively correlated with the $\theta_{LFP}$ frequency (Fig. 3h, $R = 0.29$, $P < 10^{-323}$, $t$-test for correlation coefficients, $n = 2659$ co-$\theta$ periods from 160 datasets). However, there were also many outlier data points distant from the identity line in Fig. 3h, indicating that $\theta_{Vm}$ and $\theta_{LFP}$ did not always share common frequencies. Consistent with this notion, the $\theta_{Vm}$ frequencies were, on average, slightly higher than the $\theta_{LFP}$ frequencies ($P = 8.8 \times 10^{-10}$, $t_{2,658} = -6.2$, paired $t$-test, $n = 2659$ co-$\theta$ periods). Interestingly, the differences between $\theta_{Vm}$ and $\theta_{LFP}$ frequencies during each co-$\theta$ period decreased as the mean $\theta_{Vm}$ and $\theta_{LFP}$ powers increased (Fig. 3i, $R = -0.094$, $P = 9.8 \times 10^{-7}$, $n = 2,703$ co-$\theta$ periods from 160 cells). Moreover, $\theta_{Vm}$ and $\theta_{LFP}$ were not in phase. We focused on the coherent periods, during which the difference between $\theta_{LFP}$ and $\theta_{Vm}$ frequencies was <0.01 Hz, and calculated their phase differences. Considering the $\theta_{LFP}$ propagation along the mediolateral axis described in Fig. 1j, the phase differences were plotted separately for the datasets in which the recorded cells were located medial to LFP recording sites (Fig. 3j) or vice versa (Fig. 3k). The phase difference between $\theta_{LFP}$ and $\theta_{Vm}$ was not uniformly distributed, and the $\theta_{Vm}$ of the cells medial to the LFP recording sites preceded the $\theta_{LFP}$ by 74° on average (Fig. 3j, $P = 8.1 \times 10^{-8}$, $Z = 16.0$, Rayleigh test, $n = 172$ periods from 30 cells). Anatomically reversed datasets showed the opposite result, i.e., the $\theta_{Vm}$ recorded at lateral positions to the LFP recording sites followed the $\theta_{LFP}$ by 30° on average (Fig. 3k, $P = 0.038$, $Z = 3.25$, Rayleigh test, $n = 46$ periods from 16 cells). The phase relationship between $\theta_{LFP}$ and $\theta_{Vm}$ did not differ depending on the cell locations along the radial axis in the pyramidal cell layer (Supplementary Fig. 5h, i, $P > 0.1$, $K = 1.8 \times 10^3$, Kuiper test, $n = 151$ and 74 periods from 26 and 20 deep and superficial cells, respectively).

We repeated the same series of experiments using awake, head-fixed mice (Supplementary Fig. 6a, b, $n = 22$ cells from 17 mice). Neither the durations nor the frequencies of the $\theta_{LFP}$ differed between urethane-anesthetized and awake mice (Supplementary Fig. 6c, d), indicating that type 2 $\theta$ oscillations

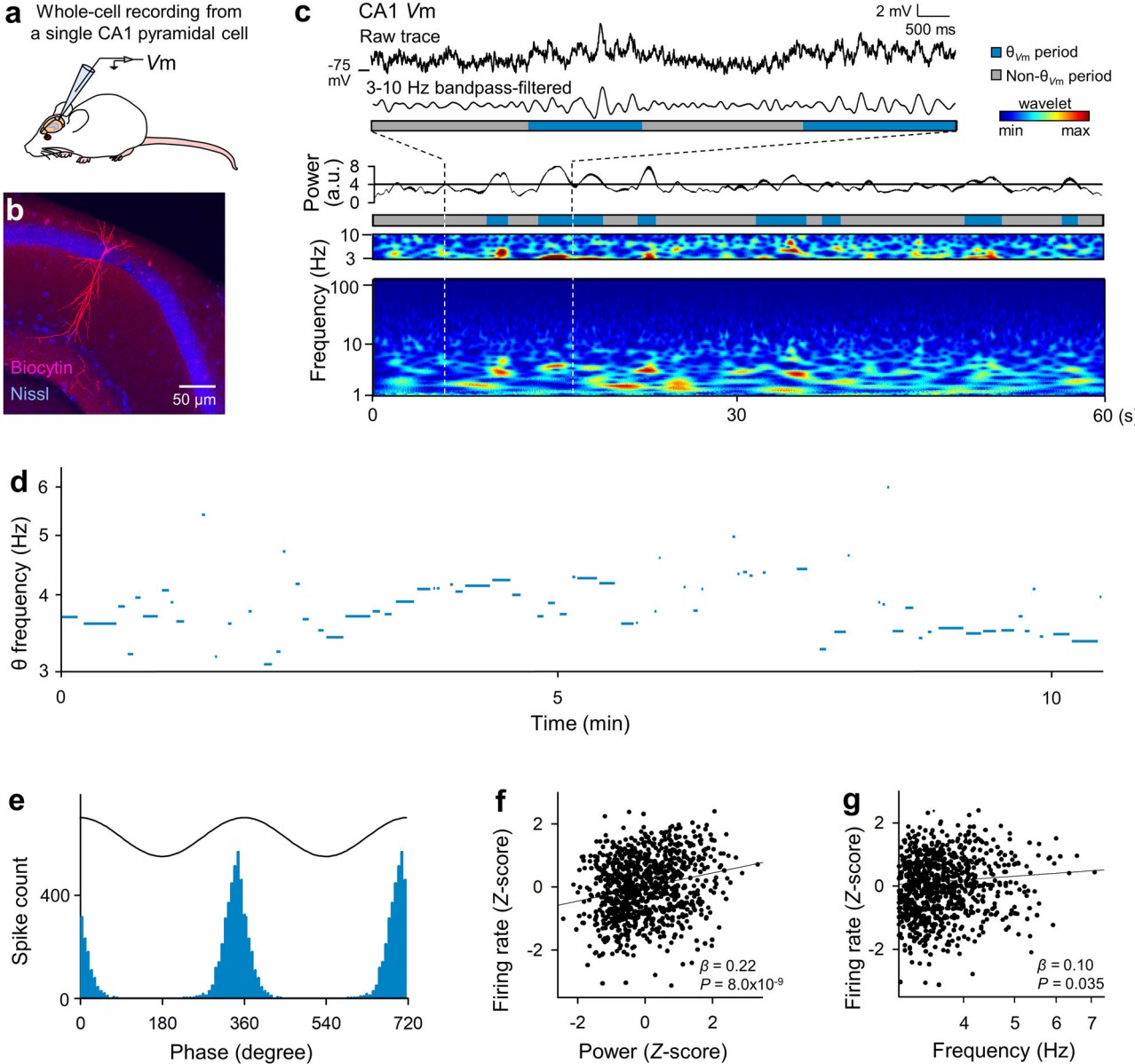

**Fig. 2 Variable θ$_{Vm}$ frequencies in a CA1 pyramidal cell. a** Schematic illustration of whole-cell current-clamp recordings from a CA1 pyramidal cell. **b** Representative confocal image of a recorded pyramidal neuron visualized with intracellular biocytin (red) and Nissl counterstain (blue). **c** A raw trace of $V$m in a CA1 pyramidal cell (top trace) was bandpass filtered between 3 and 10 Hz (second row trace) and divided into θ$_{Vm}$ periods (blue) and non-θ$_{Vm}$ periods (gray) based on the 3–10 Hz oscillation power (third row trace). The bottom plot shows the wavelet spectrogram of the $V$m trace. **d** Representative time course of the peak θ$_{Vm}$ frequencies during individual θ$_{Vm}$ periods, demonstrating that a single CA1 pyramidal cell exhibited θ$_{Vm}$ at various frequencies. **e** Cells fired spikes around the peaks of θ$_{Vm}$ cycles. $P < 10^{-323}$, $Z = 3.7 \times 10^{3}$, Rayleigh test, $n = 4655$ spikes from 220 cells. **f** The firing rates increased with increases in the θ$_{Vm}$ power. To pool data from different cells, the firing rates were $Z$-standardized on a logarithmic scale across the entire recording period of each cell. Each dot indicates the average value in a single θ$_{Vm}$ period, and the average values of the $Z$-standardized parameters are superimposed on a cell-by-cell basis. The black line indicates the line of best fit based on a generalized linear mixed model. $\beta = 0.22$, $P = 8.0 \times 10^{-9}$, $t_{1032} = 5.8$, $t$-test of the correlation coefficient, $n = 1034$ θ$_{Vm}$ periods from all 136 cells that fired at least one spike. **g** Same as **f**, but for the θ$_{Vm}$ frequencies. $\beta = 0.10$, $P = 0.035$, $t_{1032} = 2.1$, $n = 1034$ θ$_{Vm}$ periods from 136 cells.

dominated under our experimental conditions, in which mice were forced to be immobile. The data were analyzed, as shown in Fig. 3d, revealing that 10 (40%) of the 22 cells showed significant positive correlations in θ power changes (Supplementary Fig. 6e). This ratio did not differ from that of the anesthetized mice ($P = 0.71$, $\chi^2 = 0.14$, Pearson's $\chi^2$ test). The differences between θ$_{Vm}$ and θ$_{LFP}$ frequencies decreased as the mean θ$_{Vm}$ and θ$_{LFP}$ powers increased (Supplementary Fig. 6f, $\beta = -0.022$, $P = 0.0018$, $t_{2587} = -3.13$, $n = 2589$ θ periods from 22 cells).

Therefore, we concluded that θ$_{Vm}$ and θ$_{LFP}$ are partially correlated with each other and that the correlation strength depended on the oscillation states, such as θ power or θ frequency.

**Weakly correlated θ$_{Vm}$s in two CA1 pyramidal cells.** The weak θ$_{LFP}$-θ$_{Vm}$ coupling described above led to the inference that θ$_{Vm}$s of cell pairs are coherent. We obtained simultaneous patch-clamp recordings from two CA1 pyramidal cells (Supplementary Fig. 7a) and collected 125 dual patch-clamp datasets ($V$m1 and $V$m2)

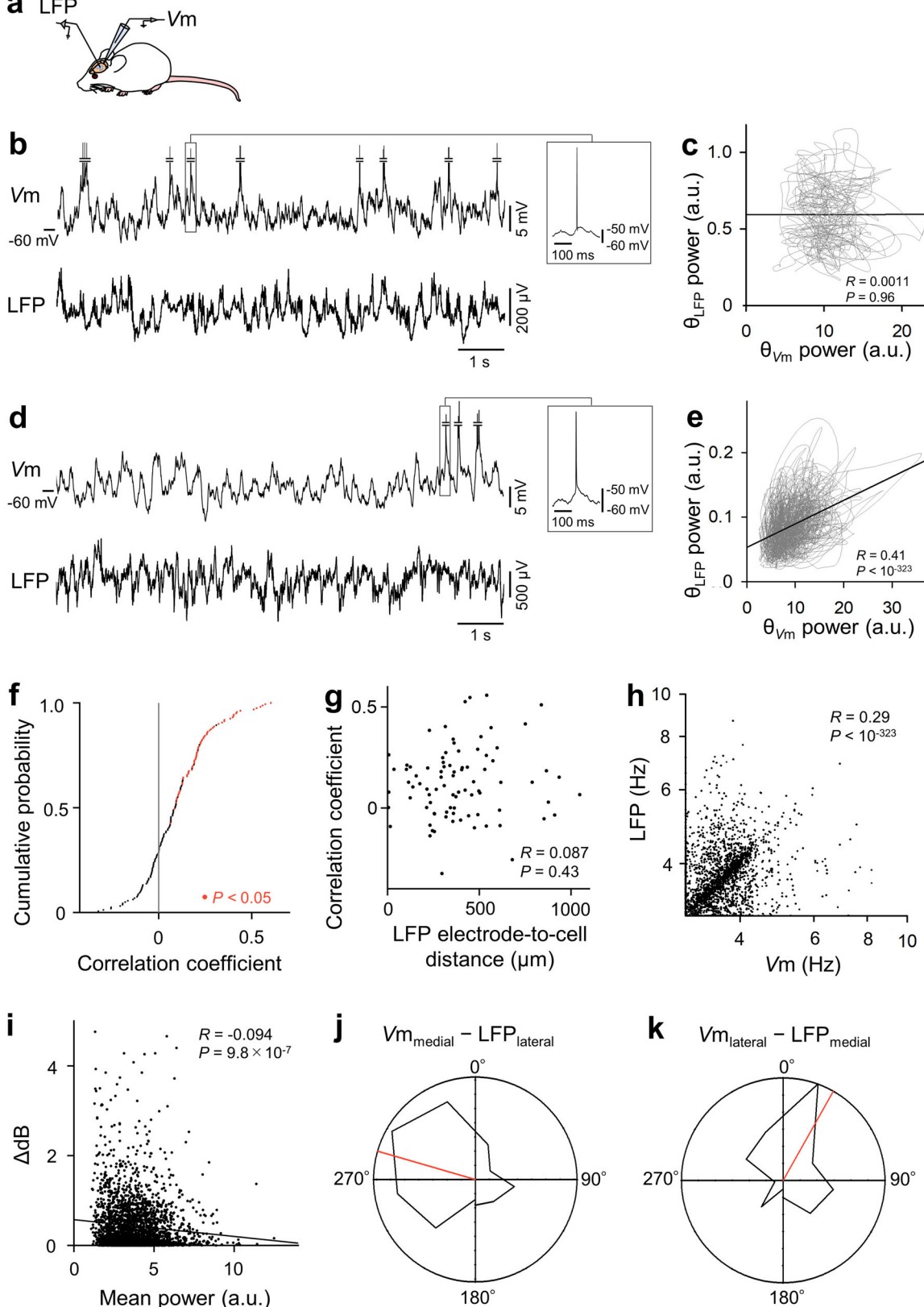

from 82 mice. The recording periods ranged from 36 s to 2097 s (median = 150 s). We applied the same analyses as in Fig. 3 to the dual patch-clamp recording datasets. We first plotted the time changes in the $\theta_{Vm}$ powers of two cells (Supplementary Fig. 7b–e) and found that 80 (64%) of the 125 cell pairs showed significant positive correlations in their $\theta_{Vm}$ power changes (Supplementary

Fig. 7c, f), whereas the remaining 45 cell pairs did not (Supplementary Fig. 7e, f). Compared with the LFP-cell pair results in Fig. 3e, the proportion of significantly correlated cell pairs was high, which might reflect the different LFP states in separate datasets. However, the proportion of cell pairs with positive correlations did not change significantly from the proportion

**Fig. 3 Weak correlations between $\theta_{LFP}$ and $\theta_{Vm}$. a** Schematic illustration of simultaneous recordings of LFPs and $Vm$ from a CA1 pyramidal cell. **b** Representative raw traces of simultaneously recorded CA1 LFPs and $Vm$s of CA1 pyramidal cells. The middle of the action potentials was omitted to enlarge the changes in $Vm$, and a full action potential for each trace in b and c is shown on the right. **c** Temporal relationships in the power of $\theta_{LFP}$ and $\theta_{Vm}$. The $\theta$ power was plotted every 100 ms over the entire recording period in each dataset. No significant correlation was observed ($R = 0.0011$, $P = 0.96$, $t$-test for correlation coefficients, $n = 1999$ 1-s segments). The black lines indicate the lines of best fit based on least-squares regression. **d, e** Same as **b, c**, but for a dataset in which a significant positive correlation was observed ($R = 0.41$, $P < 10^{-323}$, $n = 6999$ 1-s segments). **f** Cumulative probability distribution of the correlation coefficients between $\theta_{LFP}$ and $\theta_{Vm}$ powers for all 160 recorded datasets. Red dots indicate cells exhibiting significant positive correlations. **g** The correlation coefficients between the $\theta_{LFP}$ and $\theta_{Vm}$ powers (calculated in **d**) were plotted against the spatial distance between the tip of the LFP recording electrode and the patch-clamped cell. $R = 0.087$, $P = 0.43$, $t$-test for correlation coefficients, $n = $ all 86 cells whose loci were confirmed *post hoc*. **h** Relationships between the frequencies of $\theta_{LFP}$ and $\theta_{Vm}$ during co-$\theta$ periods when $\theta$ oscillations occurred simultaneously in LFPs and $Vm$. Each dot indicates a single co-$\theta$ period. $R = 0.29$, $P < 10^{-323}$, $t$-test for correlation coefficients, $n = 2659$ co-$\theta$ periods from 160 cells. **i** The difference in the $\theta$ frequencies of $\theta_{LFP}$ and $\theta_{Vm}$ in a co-$\theta$ period was negatively correlated with the geometric average of their powers. Each dot represents a co-$\theta$ period. The black line indicates the line of best fit based on least-squares regression. $R = -0.094$, $P = 9.8 \times 10^{-7}$, $n = 2702$ co-$\theta$ periods from 160 cells. **j, k** Circular distribution of the $\theta$ phase difference between LFPs and $Vm$ when $\theta_{LFP}$ and $\theta_{Vm}$ occurred simultaneously at similar frequencies ($\Delta$ frequency < 0.01 Hz). Because $\theta_{LFP}$ propagates along the mediolateral axis, the datasets were divided into two groups, in which the locations of the recorded cells were medial to the LFP recording sites (**j**) and vice versa (**k**). Red lines show the mean $\theta$ phase differences (−74° and 30° for left and right panels, respectively). The distribution was significantly nonuniform (**j** $P = 8.1 \times 10^{-8}$, $Z = 16.0$, Rayleigh test, $n = 172$ periods from 30 cells; **k** $P = 0.038$, $Z = 3.25$, Rayleigh test, $n = 46$ periods from 16 cells).

calculated for only the datasets used in Fig. 3 (Supplementary Fig. 8a, b, $P = 0.49$, $\chi^2 = 0.47$, chi-square test, $n = 125$ (all data) and 98 (data in Fig. 3 only) cell pairs, respectively). Cell pairs whose somata were physically closer had a stronger correlation (Supplementary Fig. 7g, $R = -0.24$, $P = 0.033$, $n = 78$ cell pairs). The $\theta$ frequencies during each co-$\theta$ period were significantly correlated between the two cells (Supplementary Fig. 7h, $R = 0.25$, $P < 10^{-323}$, $n = 2027$ $\theta$ periods from 125 cell pairs), although they were not fully consistent. The differences between $\theta_{Vm}$ frequencies decreased as the mean $\theta_{Vm}$ powers of the two cells increased (Supplementary Fig. 7i, $\beta = -0.011$, $P < 10^{-323}$, $t_{34,628} = -47.3$, $n = 34,630$ $\theta$ periods from 125 cell pairs). Therefore, $\theta_{Vm}$s were correlated between adjacent cells, but again, these correlations were only partial.

**Correlated $\theta$ power changes among LFPs and multiple cells.** We expanded our experiments to triple patch-clamp recordings with CA1 LFP recordings (Fig. 4a, b). We collected 21 triple-patching and LFP datasets ($Vm1$, $Vm2$, $Vm3$, and LFP) from 15 mice. The recording periods ranged from 31 s to 900 s (median = 80 s). For the sake of simplicity, we first investigated the time-dependent changes in the $\theta$ power by focusing on the pairwise correlations between LFPs and the average value of three $Vm$s (Fig. 4c–e). Of 21 datasets, 9 (43%) had significant positive correlations between the $\theta_{LFP}$ power and mean $\theta_{Vm}$ power (Fig. 4c, e), whereas the other 12 did not (Fig. 4d, e).

We next focused on the instantaneous $\theta_{LFP}$-$\theta_{Vm}$ correlations. The $\theta_{LFP}$ power became stronger when more cells simultaneously exhibited $\theta_{Vm}$s (Fig. 4f, $P = 0.0060$, $Z = 2.51$, Jonckheere trend test, $n = 13$ datasets that included at least one period during which all recordings simultaneously exhibited $\theta$ oscillations). When all three cells simultaneously exhibited $\theta_{Vm}$s, the $\theta_{LFP}$ power was positively correlated with the squared inverse of the coefficients of variance ($1/CV^2$) of the three $\theta_{Vm}$ frequencies (Fig. 4g, $R = 0.17$, $P = 0.014$, $n = 206$ co-$\theta$ periods from 13 datasets). This result indicates that the $\theta_{LFP}$ power increased when the three cells exhibited $\theta_{Vm}$s with more similar frequencies. Moreover, the difference between the $\theta_{LFP}$ frequency and the mean frequency of the three $\theta_{Vm}$s was negatively correlated with $1/CV^2$ of the three $\theta_{Vm}$ frequencies, indicating that the $\theta_{LFP}$ frequency approached the mean $\theta_{Vm}$ frequency when the three $\theta_{Vm}$ frequencies were more similar (Fig. 4h, $R = -0.78$, $P < 10^{-323}$, $t_{204} = 17.7$, $n = 206$ co-$\theta$ periods from 13 datasets).

**Machine learning-based prediction of LFPs.** These instantaneous $\theta_{LFP}$-$\theta_{Vm}$ correlations motivated us to hypothesize that the dynamics of $\theta_{LFP}$ at each moment can be estimated, at least in part, from the $\theta_{Vm}$s of three cells. To test this hypothesis, we sought to predict the $\theta_{LFP}$ waveforms from three $\theta_{Vm}$ waveforms using a machine learning model. Assuming that $\theta_{Vm}$s are associated with $\theta_{LFP}$ in a nonlinear manner, we employed a deep neural network (DNN) with a convolutional layer (Fig. 5a, Supplementary Fig. 9). Among the 21 triple patch-clamp recording datasets, we selected 8 datasets with recording periods >3 min to ensure sufficient sample sizes for training the DNN. In each dataset, the $Vm$ and LFP traces were bandpass filtered between 3 and 10 Hz, divided into 10 subsets in the recording time, and further divided into 1-s segments. We trained the DNN using 1-s segments in 9 subsets of the recorded data to predict the $\theta_{LFP}$ waveforms during 1-s segments in the remaining subset (1/10 subsets), which were not used to train the DNN (Fig. 5b, real). For each prediction, the prediction error between the original and predicted $\theta_{LFP}$ traces was quantified by the root mean square error (RMSE). We also trained the DNN using randomized pseudodata, in which 1-s segments from the training period were shuffled within each cell. We then predicted the $\theta_{LFP}$ waveform (Fig. 5b, shuffle) and computed the RMSEs for the original $\theta_{LFP}$ waveform and the $\theta_{LFP}$ waveform predicted by the shuffled data. We repeated this procedure so that all segments in the entire recording period were targeted for prediction. The RMSEs were pooled in a cumulative plot (Fig. 5c). The prediction performance was evaluated using the $D$ value of a two-sample Kolmogorov–Smirnov test; in all 8 datasets, the RMSEs of the real data were significantly lower than those of the shuffled data (Fig. 5d, 3 cells). These results indicate that the DNN predicted $\theta_{LFP}$ waveforms based on the $\theta_{Vm}$ dynamics of as few as three cells with accuracy significantly higher than chance. However, the DNN significantly predicted zero or only two of the 24 datasets based on the $\theta_{Vm}$ dynamics of only one or two cells, respectively (Fig. 5d, 1 cell, 2 cells). The $D$-values of the predictions based on the $\theta_{Vm}$ dynamics of three cells were significantly higher than those based on the $\theta_{Vm}$ dynamics of fewer cells, indicating that the $\theta_{LFP}$ dynamics reflect the collective features of multiple $\theta_{Vm}$ dynamics, which needed to be collected from at least three cells. Supplementary Fig. 10 summarizes the $D$-values and their significance in all datasets.

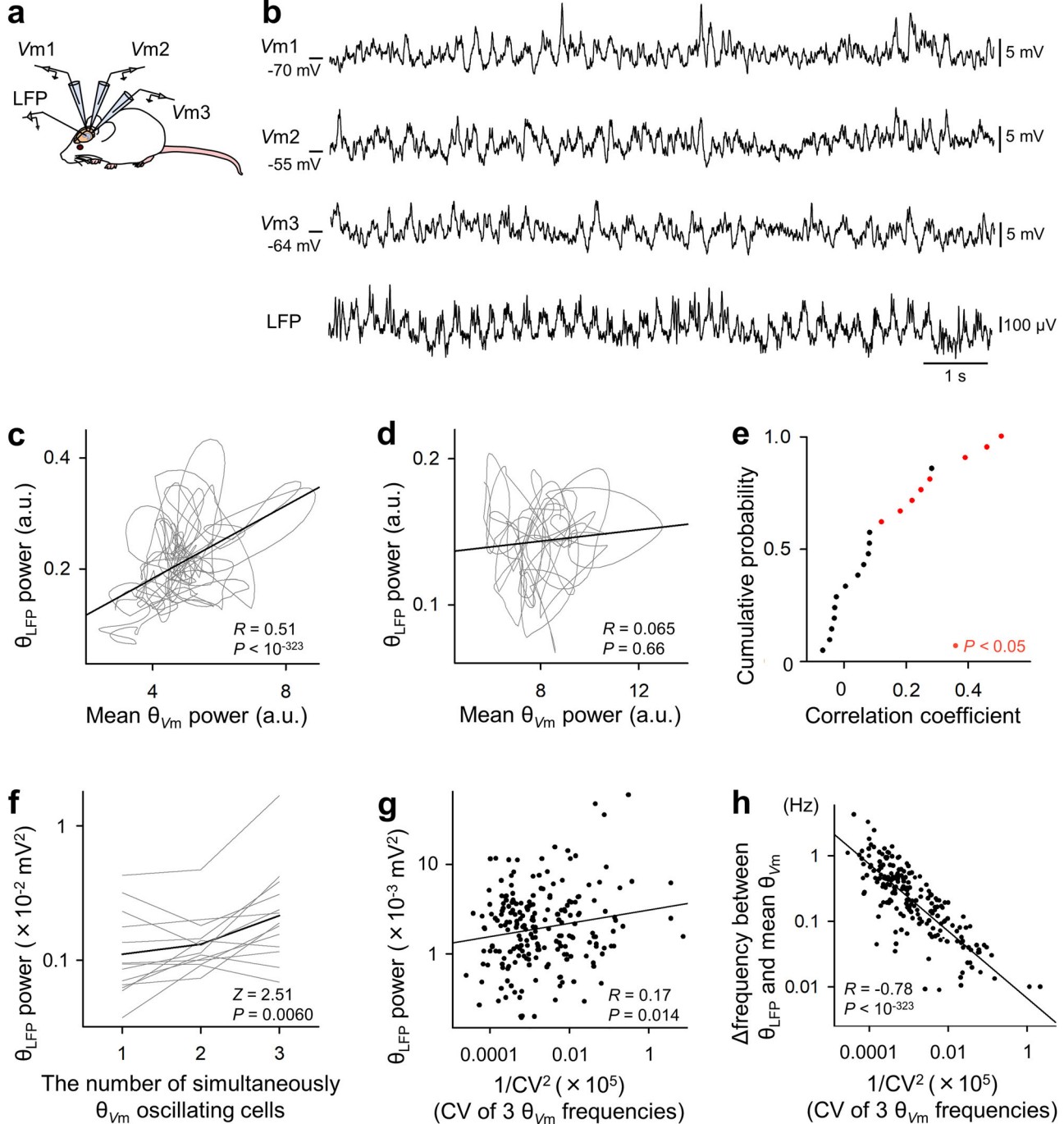

**Fig. 4 Stronger correlations of θ$_{LFP}$ for more correlated θ$_{Vm}$s. a** Schematic illustration of simultaneous recordings of LFPs and $V$ms of three CA1 pyramidal cells ($V$m1, $V$m2, and $V$m3). **b** Representative raw traces of simultaneously recorded LFPs and $V$ms of three CA1 pyramidal cells. **c, d** The θ$_{LFP}$ power during a 1-s segment plotted against the geometric mean of the θ$_{Vm}$ powers in three cells as a function of time. The triple recording in c exhibited a significant positive correlation ($R = 0.51$, $P < 10^{-323}$, t-test for correlation coefficients, $n = 740$ segments), whereas the dataset in **d** did not ($R = 0.065$, $P = 0.66$, $n = 470$ segments). The black lines indicate the lines of best fit based on least-squares regression. **e** Cumulative probability distribution of the pairwise correlation coefficients between θ$_{LFP}$ powers and the mean θ$_{Vm}$ powers of three cells for all 21 datasets. Each red dot indicates a dataset with a significant positive correlation. **f** The θ$_{LFP}$ power increased as a function of the number of cells that simultaneously emitted θ$_{Vm}$s. Each gray line indicates a single dataset, and the black line represents the mean. $P = 0.0060$, $Z = 2.51$, Jonckheere trend test, $n = 13$ datasets. **g** The θ$_{LFP}$ powers were positively correlated with the similarity of three θ$_{Vm}$ frequencies. The similarity was defined as the squared inverse of the coefficients of variance (1/CV$^2$) of three θ frequencies for co-θ periods during which LFPs and three cells simultaneously exhibited θ oscillations. Each dot indicates a single co-θ period. $R = 0.17$, $P = 0.014$, t-test for correlation coefficients, $n = 206$ co-θ periods from 13 cell triplets. The black line indicates the line of best fit based on least-squares regression. **h** The difference between the θ$_{LFP}$ frequency and the geometric mean of three θ$_{Vm}$ frequencies was negatively correlated with the similarity of three θ$_{Vm}$ frequencies. Each dot indicates a single dataset. $R = -0.78$, $P < 10^{-323}$, $n = 206$ co-θ periods from 13 cell triplets. The black line indicates the line of best fit based on least-squares regression.

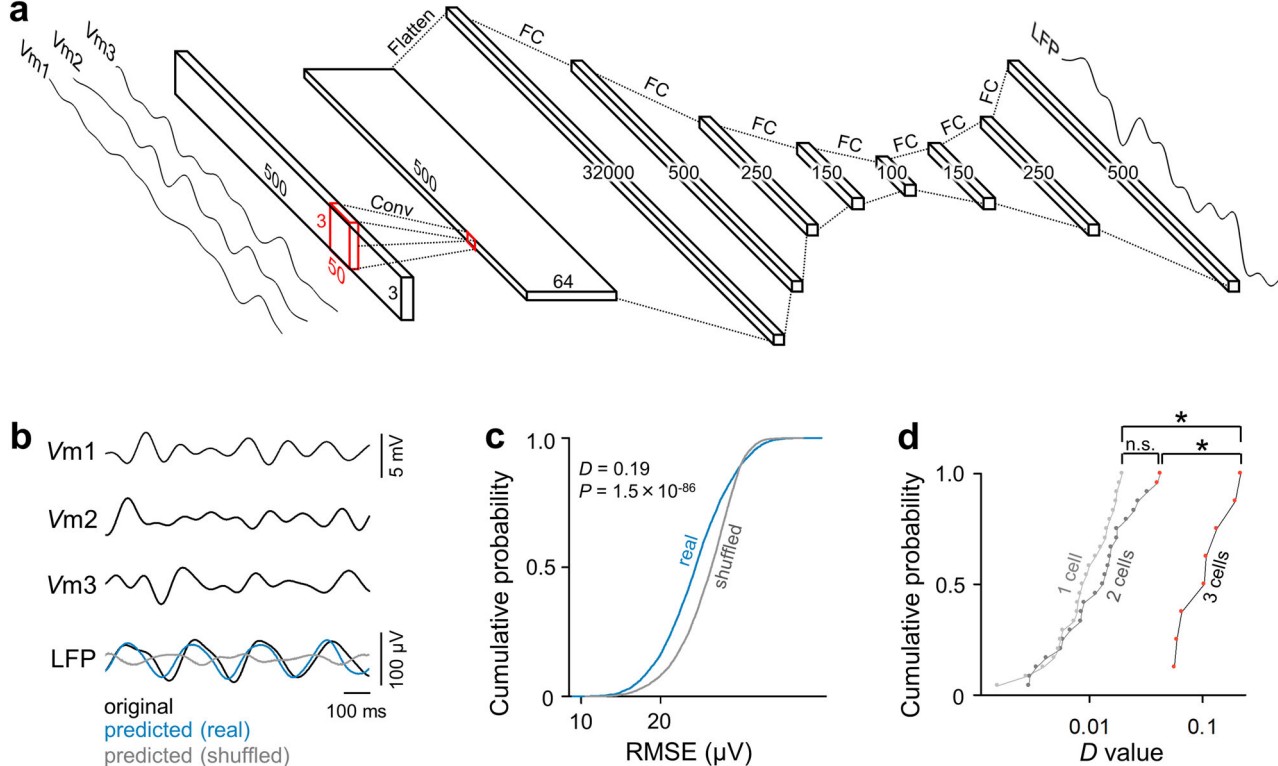

**Fig. 5 Prediction of $\theta_{LFP}$ from $\theta_{Vm}$s. a** Architecture of our neural network model. The numbers indicate the channel features (bin) in each layer. Conv: convolutional, FC: fully connected. The model input was three simultaneously recorded $V$ms ($Vm1$, $Vm2$, and $Vm3$) that were bandpass filtered between 3 and 10 Hz (left). The model was trained to output the corresponding bandpass-filtered LFPs (right). In **d**, one or two $V$ms were used as the inputs. **b** Representative traces of three $V$ms, the original LFPs (black), and the LFPs predicted from the real data (blue) or shuffled data (gray). The $V$m and the original LFP traces were bandpass filtered between 3 and 10 Hz. The shuffled data were created by randomizing the temporal order of all 1-s segments within each cell and used to train the neural network. **c** Cumulative probability distribution of the root mean squared errors (RMSEs) between the original LFP waveforms and the LFP waveforms predicted from real or shuffled data. $P = 1.5 \times 10^{-86}$, $D = 0.19$, two-sample Kolmogorov–Smirnov test, $n = 2310$ 1-s segments in a single dataset. **d** Cumulative probability distribution of the $D$-values calculated as in **c** for all 8 datasets with LFPs and $V$ms of 3 cells, as well as 24 datasets with LFPs and $V$ms of 1 or 2 cells. Each dot indicates a single dataset, and red dots indicate significant $D$-values. The LFP waveforms were better predicted from $V$ms of 3 cells than those of 1 or 2 cells. $D_{1 \; vs. \; 2 \; cells} = 0.25$, $P_{1 \; vs. \; 2 \; cells} = 0.39$, $D_{1 \; vs. \; 3 \; cells} = 1.0$, $P_{1 \; vs. \; 3 \; cells} = 2.3 \times 10^{-6}$, $D_{2 \; vs. \; 3 \; cells} = 1.0$, $P_{2 \; vs. \; 3 \; cells} = 2.3 \times 10^{-6}$, two-sample Kolmogorov–Smirnov test.

The same analysis was conducted using bandpass-filtered traces at 60–100 Hz (high gamma), 25–55 Hz (low gamma), and 0.5–1 Hz (slow oscillations). In contrast to the findings with the θ frequency band, significantly higher accuracies were obtained in only a small proportion of the datasets, regardless of the number of cells used for the prediction (Supplementary Fig. 11, high gamma: 5/24 datasets for 1 cell and 2/24 datasets for 2 cells; low gamma: 1/24 dataset for 1 cell and 2/24 datasets for 2 cells; slow oscillations, 4/24 datasets for 1 cell, 4/24 datasets for 2 cells and 2/8 datasets for 3 cells). For all three frequency bands, increases in the number of cells used for the predictions did not correspond to an increase in the prediction accuracy (Supplementary Fig. 11a, $D_{1 \; vs. \; 2 \; cells} = 0.17$, $P_{1 \; vs. \; 2 \; cells} = 0.86$, $D_{1 \; vs. \; 3 \; cells} = 0.50$, $P_{1 \; vs. \; 3 \; cells} = 0.066$, $D_{2 \; vs. \; 3 \; cells} = 0.46$, $P_{2 \; vs. \; 3 \; cells} = 0.11$; Supplementary Fig. 11b, $D_{1 \; vs. \; 2 \; cells} = 0.21$, $P_{1 \; vs. \; 2 \; cells} = 0.62$, $D_{1 \; vs. \; 3 \; cells} = 0.25$, $P_{1 \; vs. \; 3 \; cells} = 0.79$, $D_{2 \; vs. \; 3 \; cells} = 0.78$, $P_{2 \; vs. \; 3 \; cells} = 0.29$; Supplementary Fig. 11c, $D_{1 \; vs. \; 2 \; cells} = 0.46$, $P_{1 \; vs. \; 2 \; cells} = 0.0082$, $D_{1 \; vs. \; 3 \; cells} = 0.33$, $P_{1 \; vs. \; 3 \; cells} = 0.43$, $D_{2 \; vs. \; 3 \; cells} = 0.50$, $P_{2 \; vs. \; 3 \; cells} = 0.066$; two-sample Kolmogorov-Smirnov test, $n = 24$, 24, and 8 datasets for 1, 2, and 3 cells, respectively). To examine the relationships among LFPs and $V$ms in frequency bands other than θ, we calculated the cross-correlograms between pairs of bandpass-filtered LFPs, the correlation coefficients between LFP and $V$m powers of single cells, and the correlation coefficients between LFPs and the mean $V$m power of three simultaneously recorded cells for all three frequency bands. Overall,

the correlations among LFPs and $V$ms were comparable to those in the θ frequency band (Supplementary Fig. 12a–i). However, the average power spectra of the LFPs and $V$ms over the entire recording period showed that θ power dominated in all other frequency bands (Supplementary Fig. 12j, k). Therefore, the predictability of LFP traces from $V$m traces was applicable only for physiologically dominant oscillations, suggesting that the DNN could extract biologically prominent signals.

If the $\theta_{LFP}$ dynamics reflected the simple summation of multiple $\theta_{Vm}$s, the linear summations of the $\theta_{Vm}$ traces were expected to become more similar to the $\theta_{LFP}$ traces as the number of cells increased. As shown in Supplementary Fig. 13a, we analyzed the correlation between the $\theta_{LFP}$ traces and the mean $\theta_{Vm}$ traces of 1, 2, or 3 cells for each 1-s segment used in the $\theta_{LFP}$ prediction. However, contrary to expectations, the correlation coefficients decreased as the number of cells increased ($D_{1 \; vs. \; 2 \; cells} = 0.029$, $P_{1 \; vs. \; 2 \; cells} = 6.5 \times 10^{-42}$, $D_{1 \; vs. \; 3 \; cells} = 0.049$, $P_{1 \; vs. \; 3 \; cells} = 4.1 \times 10^{-59}$, $D_{2 \; vs. \; 3 \; cells} = 0.021$, $P_{2 \; vs. \; 3 \; cells} = 1.3 \times 10^{-12}$, two-sample Kolmogorov-Smirnov test, $n = 112{,}740$, 112,740, and 37,580 1-s segments from 1, 2, and 3 cells, respectively, in 8 mice). This result indicates that the mean $\theta_{Vm}$ traces of more cells were less similar to the $\theta_{LFP}$ trace. The $\theta_{LFP}$ and $\theta_{Vm}$ powers were also calculated for each 1-s segment, and the correlation between the dynamics of the $\theta_{LFP}$ power and the dynamics of the mean $\theta_{Vm}$ power of 1, 2, or

3 cells were analyzed in each dataset (Supplementary Fig. 13b). There were no significant differences among the three distributions, indicating that the similarity between the dynamics of the $\theta_{LFP}$ power and the mean $\theta_{Vm}$ power of multiple cells did not change significantly depending on the number of cells ($D_{1 \ vs. \ 2 \ cells} = 0.17$, $P_{1 \ vs. \ 2 \ cells} = 0.86$, $D_{1 \ vs. \ 3 \ cells} = 0.25$, $P_{1 \ vs. \ 3 \ cells} = 0.79$, $D_{2 \ vs. \ 3 \ cells} = 0.17$, $P_{2 \ vs. \ 3 \ cells} = 0.99$, two-sample Kolmogorov–Smirnov test, $n = 24, 24$, and 8 datasets from 1, 2, and 3 cells, respectively, in 8 mice). These results suggest that the ability to predict $\theta_{LFP}$ traces from the $\theta_{Vm}$ traces of three cells is not explained solely by the linear sum of multiple $\theta_{Vm}$s. Taken together, the DNN could predict $\theta_{LFP}$ waveforms from the $\theta_{Vm}$ dynamics of three cells based on their nonlinear relationships.

## Discussion

We investigated the temporal correlations in extracellular and intracellular θ oscillations by directly comparing LFPs with the $V$ms of multiple CA1 pyramidal cells in vivo. We found that in terms of θ powers and frequencies, the $\theta_{Vm}$s of hippocampal CA1 pyramidal cells were loosely correlated with each other and with $\theta_{LFP}$. These correlations were not explained by the anatomical locations and likely arose due to functionally coherent network activity. Consistent with this idea, we demonstrated, using a machine learning technique, that the $\theta_{LFP}$ waveforms could be predicted from the $\theta_{Vm}$ waveforms of only three pyramidal cells. Given that LFPs usually reflect the activity of considerably more than three neurons, our results suggest that many neurons simultaneously emit partially correlated $\theta_{Vm}$s and that this loose simultaneity constitutes hippocampal θ states and is experimentally captured as $\theta_{LFP}$.

We first demonstrated that the temporal evolution of $\theta_{LFP}$ power was similar across our target area in the dorsal CA1 region. Previous studies have reported that $\theta_{LFP}$ travel along the longitudinal axis of the hippocampus[11,31,32] and that $\theta_{LFP}$ phase shifts monotonically as a function of the distance along the longitudinal axis, reaching ~180° between the septal and temporal poles[31]. Consistent with a previous report[11], mediolateral propagation of the $\theta_{LFP}$ was observed in our study instead of anteroposterior propagation, despite the different animal conditions, i.e., awake and anesthetized conditions in the previous and present study, respectively. Therefore, in the present study, we considered the phase shift when analyzing the phase difference between $\theta_{LFP}$ and $\theta_{Vm}$ (Fig. 3h), and the overall transition of the $\theta_{LFP}$ power was considered synchronous within our target area. The difference in the type of $\theta_{LFP}$ between awake and anesthetized animals should also be noted in considering the functional significance of our findings. The $\theta_{LFP}$ we recorded from urethane-anesthetized mice were atropine-sensitive type 2 $\theta_{LFP}$, which usually have frequencies in the 4–7 Hz range; note that type 1 $\theta_{LFP}$ is resistant to atropine, have higher frequencies of ~8 Hz, and occur mainly during active locomotion and rapid eye movement sleep[13,14]. We also recorded $\theta_{LFP}$ in awake mice. Although we did not examine the effect of muscarinic receptor antagonists, the $\theta_{LFP}$ were also likely type 2 because the mice were immobile under head-fixed conditions, and their $\theta_{LFP}$ frequencies did not differ from those of anesthetized mice. Under anesthesia, the CA1 region may receive less input from the entorhinal cortex, but in both urethane-anesthetized and awake mice, cholinergic and GABAergic inputs from the medial septum are essential to generate θ oscillations[30,33–37]. Although type 1 and 2 $\theta_{LFP}$s have different atropine sensitivities and θ frequency ranges, they share many common features; for example, both types depend on medial septal afferents and have similar θ phase distributions in the dorsal hippocampus[33,37,38].

Therefore, we believe that the mechanisms underlying the observed cell-to-LFP or cell-to-cell correlations could be partially shared by type 1 $\theta_{LFP,}$ at least within the local hippocampal circuit. Supporting this theory, pairwise $\theta_{Vm}$ coherences during type 1 $\theta_{LFP}$ in awake, behaving mice have been reported to range from 0 to 0.8, regardless of the cell-to-cell distance[39], which is consistent with our findings on type 2 $\theta_{LFP}$. However, the anesthetized condition causes distinct activity in the medial septum[40] and the entorhinal cortex[41,42]. As a result, instructive signals for learning-dependent neuronal activity are lacking, as observed during type 1 $\theta_{LFP}$ in the hippocampus[43–46]. Future experiments recording intracellular activity from multiple hippocampal neurons in awake behaving animals should clarify the subthreshold coordination of behavior-relevant cell assemblies in $\theta_{LFP}$ state.

LFPs are shaped by collecting a myriad of electrical currents arising from neural events, such as synaptic inputs and action potentials. Therefore, intuitively, the relationship between LFPs and $V$ms might be influenced by the physical distance from the cell body. However, we did not find that the cell-to-LFP correlations in the θ power or frequency changed with the cell-to-LFP distance, at least within a radius of ~1,000 μm. One possible explanation for this discrepancy is that synaptic inputs are received by dendrites distal from the cell bodies. The dendrites of many neurons intersect, generating spatially overlapping synaptic currents. The complexity of individual current sources may blur the dependency for cell-to-LFP distances. Another but more plausible possibility is that cell-to-cell coherence arises from intrinsic cell assembly dynamics. We recorded from only three cells out of numerous neurons in the dorsal hippocampus. Even though these cells were selected in a pseudorandom and blinded manner, their $\theta_{Vm}$s were partially correlated, enough for our machine learning model to predict $\theta_{LFP}$. Therefore, we believe that the $\theta_{LFP}$-to-$\theta_{Vm}$ correlations observed in the present study did not arise from a direct causal relationship produced by the neurons we recorded but rather from the activity of other neurons located closer to the LFP recording site that behaved in a similar manner to the recorded neurons.

The coordination of $\theta_{LFP}$ and $\theta_{Vm}$s could be achieved by the interplay of inhibitory inputs from various types of interneurons[47–49] and excitatory inputs mainly from CA3 pyramidal cells under our anesthetized condition[46,50,51], each of which phase-locks to a specific $\theta_{LFP}$ phase. Among the various types of hippocampal CA1 interneurons, soma-targeting parvalbumin-positive basket cells (PV-BCs) and cholecystokinin-positive basket cells (CCK-BCs) effectively regulate $\theta_{Vm}$[37,46,52]. Considering that PV- and CCK-BCs differentially innervate CA1 pyramidal cells in the deep and superficial layers[51], i.e., PV-BCs and CCK-BCs preferentially project onto deep and superficial cells, respectively, the phase relationships between $\theta_{LFP}$ and $\theta_{Vm}$ may differ for deep and superficial cells[53]. However, Supplementary Fig. 5h indicates that deep and superficial cells showed similar $\theta_{LFP}$ phase preferences[53]. Because deep cells receive more inputs from the medial entorhinal cortex than superficial cells, this discrepancy might indicate that excitatory inputs from the medial entorhinal cortex, which are weakened under anesthesia, contribute to the shifted phase preference of deep cells. In addition, the variable holding $V$m may have affected the phase lag between the peaks of $\theta_{LFP}$ and $\theta_{Vm}$[46], while the overall phase lags were consistent with those in previous intracellular studies[37,54]. Although we did not note any anatomical bias in the relationship between $\theta_{LFP}$ and $\theta_{Vm}$, presumably due to the anesthesia and the large distance between the LFP and $V$m recording sites ($\varphi < 1,000$ μm), the temporally precise and partially shared excitatory and inhibitory inputs

among CA1 pyramidal cells may organize multicellular coordination in the $\theta_{LFP}$ state.

Using a DNN, we predicted the $\theta_{LFP}$ traces from the $\theta_{Vm}$ traces of three CA1 pyramidal neurons. Previous in silico studies have simulated LFPs based on modeled neuronal activity in the neocortex[55–57]. However, no studies to date have attempted to predict real LFPs based on the membrane potentials of multiple neurons in vivo. This lack of data is partly due to the lack of simultaneous recordings of LFPs and Vms from multiple neurons, which is technically difficult to accomplish, especially in the hippocampus. Therefore, the present study provides the first evidence that $\theta_{LFP}$ dynamics are predictable from the raw $\theta_{Vm}$ traces of three neurons in the hippocampus in vivo. It remains unclear why the predictability increased in a nonlinear manner when the number of cells used for $\theta_{LFP}$ prediction was increased from two to three. One possible explanation is composite factors among the $\theta_{Vm}$s of multiple neurons, as implied in our triple-patching datasets (Fig. 4); possible combinations among these factors could increase in a nonlinear manner as the number of cells increases. Further investigations are needed to elucidate latent factors shared across $\theta_{Vm}$s that collectively predict $\theta_{LFP}$ dynamics.

One limitation of this study is that our prediction approach did not reflect the diversity of pyramidal cells embedded in anatomically and physiologically nonuniform hippocampal circuits[51,58,59]. While our study did not observe any anatomical or physiological biases within our recorded cells (Supplementary Fig. 5), the $\theta_{Vm}$s of individual pyramidal cells should show distinct relationships with $\theta_{LFP}$ according to different input patterns and intrinsic properties in awake behaving animals[53,60]. The DNN used to predict $\theta_{LFP}$ from $\theta_{Vm}$s may be improved by considering these biased innervation and intrinsic properties. Furthermore, the prediction could benefit by including not only the $\theta_{Vm}$s of pyramidal cells but also the activity of PV- and CCK-BCs[37,46,52] as the inputs to the DNN. On the other hand, the fact that the DNN significantly predicted $\theta_{LFP}$ from $\theta_{Vm}$s of as few as three pseudorandomly selected pyramidal cells might suggest that the DNN could accommodate the spatiotemporally biased circuit structure, confirming the potential ability of the DNN to process higher-order information. Future studies should combine cell-type-specific recordings of excitatory and inhibitory neurons with computational approaches such as deep learning-based analyses to better understand the circuit mechanisms involved in coordinating network activity at the single-cell level.

## Methods

**Animal ethics**. Animal experiments were performed with the approval of the animal experiment ethics committee at the University of Tokyo (approval numbers: P29-9) and in accordance with the University of Tokyo guidelines for the care and use of laboratory animals. These experimental protocols were conducted in accordance with the Fundamental Guidelines for the Proper Conduct of Animal Experiments and Related Activities in Academic Research Institutions (Ministry of Education, Culture, Sports, Science and Technology, Notice No. 71 of 2006), the Standards for Breeding and Housing of and Pain Alleviation for Experimental Animals (Ministry of the Environment, Notice No. 88 of 2006) and the Guidelines on the Method of Animal Disposal (Prime Minister's Office, Notice No. 40 of 1995).

**Surgery and animal preparation**. Whole-cell recordings through the hippocampal window were obtained from male ICR mice (Japan SLC, Shizuoka, Japan) that were 28–45 days old[61]. Mice were anesthetized with urethane (2.25 g/kg, intraperitoneal [i.p.]). Anesthesia was confirmed by the absence of paw withdrawal, whisker movement, and eyeblink reflexes. The skin was subsequently removed from the head, and the animal was implanted with a metal head-fixation plate. A craniotomy ($2.5 \times 2.0$ mm$^2$) was performed, centered at 2.0 mm posterior to the bregma and 2.5 mm ventrolateral to the sagittal suture, and neocortical tissues above the hippocampus were aspirated[62–65]. The exposed hippocampal window was covered with 1.7% agar at a thickness of 1.5 mm. To obtain recordings from unanesthetized

mice, mice were implanted with metal head-holding plates under short-term anesthesia with 2–3% isoflurane. After full recovery, the mice received head-fixation training on a custom-made stereotaxic fixture for 1–2 h per day. The training continued for up to 5 days until the mice learned to remain calm.

**In vivo electrophysiology**. Patch-clamp recordings were obtained from neurons in the dorsal CA1 stratum pyramidale (AP: -2.0 mm; ML: 2.0 mm; DV: 1.1–1.3 mm) using borosilicate glass electrodes (4–7 MΩ). Pyramidal cells were identified by their regular spiking properties and post hoc intracellular visualization. For current-clamp recordings, the intrapipette solution consisted of the following reagents (in mM): 120 K-gluconate, 10 KCl, 10 HEPES, 10 creatine phosphate, 4 MgATP, 0.3 Na$_2$GTP, 0.2 EGTA (pH 7.3), and 0.2% biocytin. Liquid junctions were corrected offline. Cells were discarded when the mean resting potential exceeded −50 mV or the action potentials did not exceed -20 mV. LFPs were obtained from the CA1 stratum pyramidale using tungsten electrodes (UEWMGCSEKNNM, FHC, USA) coated with 1,1′-dioctadecyl-3,3,3′,3′-tetra-methylindocarbocyanine (DiI). The tungsten electrode location was detected by post hoc observation of fluorescent DiI tracks. Simultaneous quadruple LFP recordings were obtained by using four glass electrodes (0.5-2.5 MΩ) filled with artificial cerebrospinal fluid (aCSF), including DiI (4% w/v), Evans Blue (2%), or Trypan Blue (2%), for post hoc identification of the electrode locations (Fig. 1). The electrode locations estimated during the stereotaxic experiments were approximately the same as the post hoc visualized positions. Only the LFPs recorded in the pyramidal cell layer were included in the analyses. The signals from the four glass electrodes were amplified using a MultiClamp 700B amplifier, whereas the signals from the tungsten electrode were amplified using a DAM80 AC differential amplifier (World Precision Instruments). All signals were digitized at a sampling rate of 20 kHz using a Digidata 1440 A digitizer (Molecular Devices) that was controlled by pCLAMP 10.3 software (Molecular Devices). To confirm the type of $\theta_{LFP}$ (Supplementary Fig. 1), atropine was intraperitoneally injected at a dose of 50 mg/kg at least 120 s after the beginning of the LFP recordings. The effect of atropine was examined 960–1,080 s after the injection (a 120-s period).

**Histology**. Following each experiment, the electrode was carefully withdrawn from the hippocampus. The mice were transcardially perfused with 4% paraformaldehyde followed by overnight postfixation. The brains were sagittally sectioned at a thickness of 100 μm using a vibratome. The sections were incubated with 2 μg/ml streptavidin-Alexa Fluor 594 conjugate and 0.2% Triton X-100 for 4 h, followed by incubation with 0.4% NeuroTrace 435/455 blue fluorescent Nissl stain (Thermo Fisher Scientific; N21479) for 4 h. The tracks of the LFP electrodes were also detectable via DiI fluorescence. Fluorescence images were acquired using an FV1200 confocal microscope (Olympus, Tokyo, Japan) and subsequently merged. The depth of the soma and the tips of the tungsten electrodes were estimated in the Z-scan series of an FV1200 confocal microscope. More specifically, the CA1 area was roughly divided into three subareas along the proximodistal axis, and the locations of the soma and the tips of tungsten electrodes were visually classified as one of the following five positions: CA1a, the border between CA1a and b, CA1b, the border between CA1b and c, or CA1c. Then, the XYZ coordinates of the soma and the tips of tungsten electrodes in the brain were determined based on the origin located at the bregma, with the X- and Y-axes representing the mediolateral and anteroposterior axes, respectively.

**Statistics and reproducibility**. Data analyses were performed using MATLAB (R2017b, Natick, Massachusetts, USA), and the summarized data are reported as the mean ± SD unless otherwise specified. $P < 0.05$ was considered statistically significant. Sample size (the number of triple-patching datasets and single whole-cell recordings in awake condition) was determined by referencing previous publications (Jouhanneau et al., 2018, 2019). No statistical methods were used to predetermine the sample size. All conclusions of this study are based on recordings of populations of cells.

**Data analysis**. To detect periods with $\theta_{LFP}$, wavelet transformations were first conducted for each LFP recording, the sampling rate of which was reduced to 500 Hz. Any period was defined as a $\theta_{LFP}$ period if the mean absolute value of its wavelet coefficients between 3–10 Hz exceeded the mean ± 2 SDs of the values between 1–100 Hz. When the duration of $\theta_{LFP}$ or non-$\theta_{LFP}$ periods was <1 s, we regarded the duration as a non-$\theta_{LFP}$ or $\theta_{LFP}$ period, respectively. The 2 SD threshold for detecting the $\theta_{LFP}$ period was determined after ascertaining that the similarity of $\theta_{LFP}$ periods in the four simultaneously recorded LFPs at this threshold was most distinguishable from the surrogate data (Supplementary Fig. 2). $\theta_{Vm}$ periods were also detected using the same threshold. The similarity of $\theta_{LFP}$ periods in simultaneously recorded LFP pairs was quantified using the Dice similarity coefficient, the double union of two independent sets divided by the sum of the two sets. The surrogate data were created by randomly shuffling the timings of the individual $\theta_{LFP}$ periods without changing the duration of each period. The peak frequency was calculated based on the wavelet power as the absolute value of the wavelet coefficient. The wavelet power at each frequency was averaged across

the target period, and the frequency at which the mean wavelet power reached its maximum value was calculated. The $Z$ score powers of $\theta_{LFP}$ and $\theta_{Vm}$ were also calculated based on the wavelet power. We averaged the wavelet power for each frequency across 1-s bins. The maximum power for each bin and the $Z$ scores of all bins were subsequently calculated.

**Prediction of LFPs from Vms**. A custom DNN model was constructed to predict LFPs from up to three simultaneously recorded *V*ms using Python. Our DNN had an encoder-decoder structure. The encoder compressed the input (i.e., 1, 2, or 3 *V*ms) to a lower dimensional representation and extracted features from the input, whereas the decoder reconstructed the final output (i.e., LFP) from the compressed vector. In our study, the *V*ms of 1, 2, or 3 cells over 1-s windows (size 500, 2-ms bins) were first passed through a convolutional layer. In this operation, the input was transformed into a feature matrix of reduced size through convolution by kernel matrices. This kernel processing enabled the extraction of meaningful features from the input into a smaller number of parameters[66]. Then, the layer output was flattened to a one-dimensional vector (size 32,000). The one-dimensional vector was passed through four fully connected layers and finally compressed to a size of 100. This lower dimensional representation was reconstructed to a size of 500 via three fully connected layers. Our DNN was implemented using the Python deep learning library Keras and the TensorFlow backend. The network was optimized by adaptive moment estimation (Adam) with a learning rate of 0.001. The parameters for the optimizer Adam were as follows: β1 (an exponential decay rate for the first moment estimates) = 0.9, β2 (an exponential decay rate for the second moment estimates) = 0.999, ε = $10^{-7}$, and decay = 0.0001; the default values were used for the other parameters. Because our focus was θ oscillations, raw LFP and *V*m traces (20,000 Hz) were downsampled to 500 Hz and bandpass filtered between 3 and 10 Hz in Fig. 5. In Supplementary Fig. 7, the downsampled traces were bandpass filtered at 60–100 Hz, 25–55 Hz, and 0.5–1 Hz for high gamma, low gamma, and slow oscillations, respectively. Datasets with recording durations >3 min were used for prediction ($n = 8$). Our DNN model was trained to produce 1-s θ-filtered LFPs (500 bins) from up to three *V*ms of the corresponding time bins. To assess the model performance on the entire dataset, 10-fold cross-validations were used. Each dataset was equally divided into 10 subsets, and during each training session, one subset was used as test data, while the remaining 9 subsets were used as training data. To obtain sufficient numbers of training data, 1-s segments (500 bins) were extracted by shifting a 1-s time window at a step of 2 ms (1 bin) across the training data. For test data, 1-s segments were extracted by shifting a 1-s time window at a step of 100 ms. For each cross-validation, the training lasted 50 epochs with a batch size of 256, and the RMSEs were calculated to assess how well the model predicted new data that had not been used for training. As a randomized control, surrogate data were produced by shuffling the combinations of three *V*ms; that is, the time labels for segments were exchanged within each cell. The DNN was also trained using the shuffled data, and RMSEs were calculated. To evaluate the significance for prediction, the RMSEs of all 1-s segments in all 10 subsets were pooled for either original or shuffled data in each dataset, and a two-sample Kolmogorov–Smirnov test was used to calculate the *D*-values and *P*-values for each dataset.

**Model and parameter tuning**. The model architecture and training parameters were optimized on a different dataset before analyzing the main dataset. First, we compared four model structures with distinct characteristics (Supplementary Fig. 9a). In Model 1, which became our final model, the input is first passed to a convolutional layer and then to a series of fully connected layers. The structure of Model 2 is similar to that of Model 1, but without the first convolutional layer. Model 3 has only one fully connected layer. Finally, Model 4 has a convolutional layer and fully connected layers but does not have an encoder-decoder structure.

All models were trained on data that were specially prepared for model tuning (tuning datasets). Model 1, which was chosen as the final architecture in this work, showed the lowest RMSE value among all models (Supplementary Fig. 9b, c). In our model, the convolutional layer learns filters in the temporal dimension. Since convolutional layers are used to extract meaningful temporal structures, it is possible that convolutional layers successfully extracted local oscillations in the input *V*ms, which were important for predicting LFPs. The convolutional layer was followed by a series of deep layers. It is important to note that these fully connected layers have encoding and decoding architecture. By implementing this feature, the model is forced to extract and learn only the important features for predicting LFPs.

After the model architecture was selected, the model parameters were optimized based on the performance of the tuning data. As representative data, the results from four sets of parameters are shown in Supplementary Fig. 9d, e. Dropout rate of the dropout layers and learning rate were also optimized to 0.5 and 0.01 so that the RMSE takes the minimum value; the RMSE was 0.0203, 0.0229, 0.0205 when dropout rate was 0.5, 0.7, 0.9, and it was 0.0194, 0.112, 0.0201 when learning rate was 0.01, 0.001, 0.1. The optimal number of epochs was determined similarly based on the learning curve. The average of all traces shows that the RMSE value of the validation data hit the lowest at 5 epochs. After that, overfitting was observed as the RMSE of the validation data began to increase while the value of the training data continued to decrease.

**Reporting summary**. Further information on research design is available in the Nature Portfolio Reporting Summary linked to this article.

## Data availability
Source data for the presented figures are provided as Supplementary Data with this paper. The datasets generated during and/or analyzed during the current study are available from the corresponding author on reasonable request.

## Code availability
All offline computational analyses except Fig. 5 and Supplementary Fig. 9 were performed using MATLAB R2017b. The DNN based analyses in Fig. 5 and Supplementary Fig. 9 were performed using Python 3.9.1. Each analysis procedure is described in necessary detail in the Method section for others to execute. Any analysis code and analyzed data are available by sending a request to the corresponding author.

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

## Acknowledgements

This work was supported by JST ERATO (JPMJER1801), the Institute for AI and Beyond of the University of Tokyo, and JSPS Grants-in-Aid for Scientific Research (18H05525, 20K15926).

## Author contributions

A.N. and Y.I. conceptualized the study; A.N., K.Y. and N.M. performed the experiments and data analyses; A.N. and Y.I. wrote the original draft; and A.N., K.Y., N.M., and Y.I. reviewed and edited the final manuscript.

## Competing interests

The authors declare no competing interests.
