## [Peer Review File · Communications Biology]

Reviewers' comments:

Reviewer #1 (Remarks to the Author):

The paper 'Hippocampal theta oscillations represent collective dynamics of multineuronal membrane potentials' by Noguchi et al. investigates the relationship between intracellular recordings and hippocampal LFP-recordings. The issue addressed in the paper is a longstanding one and the conclusions of the authors are not entirely novel. This being said, however, it appears that the current work is well worthy of publication. The reason for this positive assessment are: (i) the issue addressed here is of overall significance; (ii) the exceptional quality of the data; (iii) the fact that the data presented here are difficult to obtain; (iv) the authors analyzed their data in novel ways. With all that I am supportive of publication. My comments are minor and concern the presentation of the data.

Minor points:

Figure 1 B provide orientation

Figure 1 C increase brightness

Figure 3 B, C show only one example spike in full; then clip spikes and massively increase Y-axis of both the Vm and the LFP trace.

Figure 3 E: Perhaps skip or at least dash the regression line, as there no relation.

Figure 4 B: Amazing data. There should be more space for this panel, perhaps shrink the schematic in panel A and increase the Y-axis such that the data can be better inspected.

Reviewer #2 (Remarks to the Author):

In this paper, Noguchi et al., examine the concerted membrane potential dynamics of individual pyramidal cells of the dorsal hippocampus and how does it relate to the local field potentials. The authors combine multiple patch clamp recordings (up to 3 cells simultaneously) and found that correlation with LFP improves as more neurons are pooled together (using a simple neural network model). While this is relatively obvious, there is something interesting in this paper. I have however many comments and concerns that require additional revision.

1- Evaluation of coordinated theta activity across LFP recordings is critical to establish the limits for cell analysis. Results show relatively high values of synchronization, which is taken as a reference for evaluating neuron-LFP synchrony later on the paper. It is unclear however how comparable are the laminar location of extracellular recordings. Can authors provide some reference (e.g. do ripple power from different channels correlate with the correlation coefficient between pairs?). How comparable are LFP locations for the intra-LFP analysis to those used in LFP-LFP analysis? Is there any rostrocaudal, mediolateral effect?

2- Provided LFP locations are comparable, there are some differences between pairs of electrodes (correlation coefficient from 0.5 to near 1.0), suggesting that comparison between cells and LFP may be confounded. First, correlation analysis of LFP signals in Fig.1 is implemented for the entire recording (and hence higher correlations are favored by the slower components). Consistently, when theta periods are more strictly selected (larger SD thresholds) correlations drop (sup.fig.2). Oddly, when similar criteria applies to intracellular signals an indirect measure of correlation rises instead (differences between the intra and extra theta frequencies becomes smaller) (Fig.3g). Thus, results appear very sensitive to the definition of the analysis windows and the location of the LFP channel. The authors should find a way to standardize comparisons.

3- Results from a neural network model to predict LFP from intracellular signals should be better supported. First, why this architecture (a convolutional input layer plus deep layers)? No hyper-parameter assessment is provided. No test for overfitting. Second, how much do results depend on

the criteria discussed above? The authors provide some hints in Supp.fig.7 by reporting prediction per oscillatory bands, but frequency band correlation was not shown for LFP-LFP and cell-LFP as a reference. Finally, that 3 cells provide better fit as compared to 1 or 2 cells seems relatively obvious. The control analysis in Sup.fig.8a appears a bit marginal.

4- A main problem of the paper is the approach. Using just 3 cells plus a distant LFP make results too much dependent on variability (recording duration, location, cell identity, local vs global influences, ensembles, etc...). The manuscript will benefit from accepting and discussing these limitations. For example, recent data support the idea that cell-type specificity is a major factor influencing contribution to LFP signals due to biases in connectivity and the definition of local microcircuits. The authors refer to basket cell interneurons in a very general way, but PV and CCK differently connect to deep and superficial CA1 cells and so their contribution to LFP may differ (as already shown for sharp-wave ripples for instance, but also during theta). Similarly, cells can join dynamically to different ensembles and therefore the ability to predict distant and local LFP should also reflect these biases.

5- There is some asymmetry in the LFP-LFP cross-correlation function (Fig.1h). Is there any mediolateral, or rostral-caudal trend?

6- Fig.1e shows a long LFP recording period, while 2C,D is shorter. Recording duration is limited for intracellular recordings, and so cell-LFP analysis may be influenced by this factor. Can the authors provide some equivalent assessment of LFP-LFP correlations?

7- In general, discussion is not updated with recent hippocampal data on intracellular versus extracellular dynamics (e.g. results regarding different theta frequencies and phase shifts between intra and extracellular signals, the role of cell-type specificity, proximodistal shifts, etc...). There are many more references to cortical dynamics than to the specific dorsal hippocampus itself.

Reviewer #3 (Remarks to the Author):

The manuscript by Noguchi et al., examines the relationship between the activity of hippocampal CA1 neurons and theta oscillations recorded in the local field potential. This study focuses on the atropine sensitive type 2 theta oscillations and demonstrates a complex relationship between individual neurons and LFP theta, whereby one population of neurons (approx. 40%) is highly correlated with LFP theta power, while the remaining neurons show no significant correlation. The authors go further by characterizing the phase relationship between VM and LFP theta oscillations, finding that even with as few as three simultaneously recorded neurons, a neural network model is able to accurately predict the amplitude and phase of extracellularly recorded theta LFPs. Clearly, the authors have invested a significant amount of time and expertise in gathering and analyzing this extensive dataset. On the whole the results are solid and presented clearly throughout, with the conclusions drawn by the authors being strongly supported by the underlying data. I would be happy to support publication of the manuscript and have included some suggestions below, which I feel would further clarify and strengthen the authors conclusions.

1. My one major concern which I feel could be easily addressed by altering the analysis a little is the potential for over and undersampling of specific cells in some of the figures presented. As far as I understand the data shown in Figures 2f & g for example include all VM theta periods (100ms each, with each dot representing a single one). This would mean that with approx. 30 % of the time being a theta state, the shortest recorded cell (30s) would contribute about 300 data points, while the longest recorded cell (2000s) would add 20000 data points. Depending on the distribution of the recording times, this has the potential to greatly skew certain results to a handful of neurons at the upper end. To overcome this, the z-scored power for each cell could be binned, with only the average power and firing rate for each bin added on a cell by cell basis.

2. The methods section could be expanded a little. There seems to be no mention of how the awake recordings differed from the anesthetized recordings, how were the animals forced to be immobile in the awake state? This should be clarified.

3. Though the authors suggest in the discussion that the same principles may apply to type I theta as observed here, it should be noted that the increased amplitude and frequency of type I theta (which the authors show in this study, both increase the correlation strength between Θ LFP and Θ Vm) may result in a larger fraction of the neuronal population being coupled to LFP theta.

4. One interesting avenue to explore would be to try and determine what differentiates the 41% of neurons that significantly correlate with LFP theta power, from the remaining 59% that don't. The authors could compare mean firing rate, resting membrane potential or potentially anatomical locations of the two groups of neurons (such as superficial vs deep CA1 cells) to establish if there is a consistent factor that determines whether or not a cell is actively engaged with the ongoing theta oscillation.

5. Since correlation strength between Θ LFP and Θ Vm is modulated by theta power and frequency, it might be prudent to rule out any changes in LFP theta. During the dual patch experiments there appears to be an increased percentage (64%), vs the 41% (single patch) and 43% (triple patch) of cells that show an increased correlation between Vm and LFP theta power. While I doubt that any systematic changes in LFP occurred specifically during the dual patch experiments, it would be better to rule out this as a potential problem.

6. In the discussion the authors touch on the role of PV-positive basket cells in mediating the association of Θ LFP and Θ Vm. While clearly beyond the scope of these data, I was wondering if they could at least discuss their thoughts on how including data from PV+ neurons might impacted the performance of their DNN.

Reviewer #1

The paper 'Hippocampal theta oscillations represent collective dynamics of multineuronal membrane potentials' by Noguchi et al. investigates the relationship between intracellular recordings and hippocampal LFP-recordings. The issue addressed in the paper is a longstanding one and the conclusions of the authors are not entirely novel. This being said, however, it appears that the current work is well worthy of publication. The reason for this positive assessment are: (i) the issue addressed here is of overall significance; (ii) the exceptional quality of the data; (iii) the fact that the data presented here are difficult to obtain; (iv) the authors analyzed their data in novel ways. With all that I am supportive of publication. My comments are minor and concern the presentation of the data.

Thank you for your positive evaluation, which encouraged us to resubmit this manuscript. We have revised our manuscript in accordance with your comments. Our point-by-point responses are as follows:

Minor points:

Figure 1 B provide orientation

Thank you for pointing out the lack of information. We added arrows indicating the glass pipette insertion directions and revised the figure legend accordingly (Fig. 1b; L. 827).

Figure 1 C increase brightness

We appreciate the comment and apologize for any visibility issues. We increased the brightness of the image (Fig. 1c).

Fig. 1: Synchronized θ_{LFPs} across the dorsal hippocampal CA1 area.

(a) Schematic illustration of simultaneous *in vivo* LFP recordings from four sites in the dorsal hippocampal CA1 area.

(b) Representative top view schematic of the hippocampus (yellow), hippocampal window (square), recording sites (black dots), and directions of the inserted glass pipettes used to acquire the data shown in d.

(c) Fluorescence image of the track of an LFP electrode visualized by DiI (red). The histological section was counterstained with fluorescent Nissl stain (blue).

Figure 3 B, C show only one example spike in full; then clip spikes and massively

increase Y-axis of both the V_m and the LFP trace.

Thank you for your helpful comment. We revised the figures so that the middle of the spikes was omitted from the V_m traces and the Y-axes of the V_m and LFP traces were expanded. An example trace including only one spike for each V_m trace is shown on the right (Fig. 3b, c; LL. 886-887).

Fig. 3: Weak correlations between θ_{LFP} and θ_{V_m} .

(b, c) Left: representative raw traces of simultaneously recorded CA1 LFPs and V_m s of CA1 pyramidal cells. The middle of the action potentials was omitted to enlarge the changes in V_m , and a full action potential for each trace in b and c is shown on the right. Right: Temporal relationships in the power of θ_{LFP} and θ_{V_m} . The θ power was plotted every 100 ms over the entire recording period in each dataset. No significant correlation was observed in the cell shown in b ($R = 0.0011$, $P = 0.96$, t -test for correlation coefficients, $n = 1,999$ time segments), whereas a significant positive correlation was observed in c ($R = 0.41$, $P < 10^{-323}$, $n = 6,999$ segments). The black lines indicate the lines of best fit based on least-squares regression.

Figure 3 E: Perhaps skip or at least dash the regression line, as there no relation.

Thank you for the comment. We removed the regression line from Fig. 3e.

Fig. 3: Weak correlations between θ_{LFP} and θ_{V_m} .

(e) The correlation coefficients between the θ_{LFP} and θ_{V_m} powers (calculated in d) were plotted against the spatial distance between the tip of the LFP recording electrode and the patch-clamped cell. $R = 0.087$, $P = 0.43$, t -test for correlation coefficients, $n =$ all 86 cells

whose loci were confirmed *post hoc*.

Figure 4 B: Amazing data. There should be more space for this panel, perhaps shrink the schematic in panel A and increase the Y-axis such that the data can be better inspected.

Thank you for acknowledging the value of the data. We decreased the size of Panel a and enlarged the traces in Panel b (Fig. 4a, b).

Fig. 4: Stronger correlations of θ_{LFP} for more correlated θ_{Vm} s.

(a) Schematic illustration of simultaneous recordings of LFPs and V_m s of three CA1 pyramidal cells (V_{m1} , V_{m2} , and V_{m3}).

(b) Representative raw traces of simultaneously recorded LFPs and V_m s of three CA1 pyramidal cells.

Reviewer #2

In this paper, Noguhi et al., examine the concerted membrane potential dynamics of individual pyramidal cells of the dorsal hippocampus and how does it relate to the local field potentials. The authors combine multiple patch clamp recordings (up to 3 cells simultaneously) and found that correlation with LFP improves as more neurons are pooled together (using a simple neural network model). While this is relatively obvious, there is something interesting in this paper. I have however many comments and concerns that require additional revision.

We thank you for these constructive comments, which led to further discoveries and greatly improved our work. Our individual responses are provided below:

1- Evaluation of coordinated theta activity across LFP recordings is critical to establish the limits for cell analysis. Results show relatively high values of synchronization, which is taken as a reference for evaluating neuron-LFP synchrony later on the paper. It is unclear however how comparable are the laminar location of extracellular recordings. Can authors provide some reference (e.g. do ripple power from different channels correlate with the correlation coefficient between pairs?). How comparable are LFP locations for the intra-LFP

analysis to those used in LFP-LFP analysis? Is there any rostrocaudal, mediolateral effect?

Thank you for noting these critical points. Regarding the laminar location of the LFP recording sites, we conducted single channel recordings at each recording site by using a glass pipette or a tungsten electrode and histologically confirmed the tip location (LL. 509-513). Recordings from layers other than the pyramidal cell layer were removed from the LFP analyses. Because of the single channel recordings, we cannot obtain precise laminar information by analyzing ripples. We will use multichannel recordings in future experiments to assess the laminar profile of the LFPs (Liu et al., 2022). We added a description of the histology-based exclusion of the LFP data from the analyses to the Methods section (LL. 515-516).

To compare the results at different LFP locations in the intra-LFP and LFP-LFP analyses, we can assume that the target area of the former is included in the area of the latter. This assumption is valid because the coordinates of the hippocampal window were consistent across all experiments, and the angle of the LFP electrodes was more vertical in the LFP-LFP recordings ($\sim 20^\circ$ from the vertical) than in the intra-LFP recordings ($\sim 40^\circ$ from the vertical); thus, the LFP electrodes for the intra-LFP recordings cannot be outside the accessible area of the LFP-LFP recordings. Although we cannot precisely compare recording sites between animals, the coordination of the θ_{LFPs} quantified in our LFP-LFP experiments could be referenced in the intra-LFP analyses. In support of this view, we showed consistent mediolateral effects in the LFP-LFP and intra-LFP data, which were added as Fig. 1h-j and Fig. 3h, respectively. In Fig. 1h and i, we calculated the cross-correlations of LFP trace pairs that were bandpass-filtered at 3-10 Hz. The time lags were calculated by referencing the LFPs recorded from relatively medial (Fig. 1h) or anterior (Fig. 1i) locations for individual pairs. The θ_{LFPs} recorded from relatively medial positions significantly preceded their counterparts (Fig. 1j ML, $P = 0.0023$ vs. time 0, $t_{14} = -3.7$, Student's t -test, $n = 15$ pairs from 5 mice), while no such effect was observed between θ_{LFPs} along the anteroposterior axis (Fig. 1j AP, $P = 0.095$ vs. time 0, $t_{13} = 1.8$, Student's t -test, $n = 14$ pairs from 5 mice), indicating the mediolateral propagation of θ_{LFP} as described by Lubenov and Siapas, 2003. We added these descriptions to the Results section (LL. 91-93, 114-126), Discussion section (LL. 368-375), and figure legends (LL. 845-857).

Fig. 1: Synchronized θ_{LFPs} across the dorsal hippocampal CA1 area.

(h) Cross-correlograms of the 14 pairs of simultaneously recorded LFP traces bandpass filtered at 3-10 Hz (gray line) and their mean (black). Only the pairs of LFPs for which

the relative positions of the electrodes along the mediolateral axis could be identified were included. In the inset, the time scale is expanded near 0 ms, and the plot indicates that the θ_{LFP} s recorded from relatively medial recording sites preceded their counterparts.

(i) Same as h, but for the 15 pairs of LFPs for which the relative positions of the electrodes along the anteroposterior axis could be identified. Cross-correlations peaked at 0-ms time lags.

(j) Time lags between pairs of θ_{LFP} s were calculated for the mediolateral (ML) and anteroposterior (AP) pairs shown in h and i, respectively. θ_{LFP} propagation was observed in the medial to lateral direction ($P = 0.0023$ vs. time 0, $t_{14} = -3.7$, Student's t -test, $n = 15$ pairs from 5 mice) but not along the anteroposterior axis ($P = 0.095$ vs. time 0, $t_{13} = 1.8$, Student's t -test, $n = 14$ pairs from 5 mice).

Fig. 3h was revised by considering the θ_{LFP} propagation along the mediolateral axis. Here, we calculated the phase difference between θ_{LFP} and θ_{V_m} by focusing on "coherent" periods in which the difference between the θ_{LFP} and θ_{V_m} frequencies was less than 0.01 Hz. The phase differences were plotted separately for the datasets in which the recorded cells were located medial to the LFP recording sites (Fig. 3h left) or vice versa (Fig. 3h right). As a result, the phase difference between θ_{LFP} and θ_{V_m} was not uniformly distributed, and the θ_{V_m} of cells at medial positions to the LFP recording sites preceded θ_{LFP} by 74° on average (Fig. 3h left, $P = 8.1 \times 10^{-8}$, $Z = 16.0$, Rayleigh test, $n = 172$ periods from 30 cells). Anatomically reversed datasets showed the opposite result, *i.e.*, the θ_{V_m} recorded at lateral positions to the LFP trailed the θ_{LFP} by 30° on average (Fig. 3h right, $P = 0.038$, $Z = 3.25$, Rayleigh test, $n = 46$ periods from 16 cells). These results may reflect the mediolateral propagation of the θ_{LFP} and are consistent with the results for the LFP-LFP data added in Fig. 1h-j. We added the description to the Results section (LL. 203-212), Discussion section (LL. 376-378), and figure legends (LL. 909-916).

Fig. 3: Weak correlations between θ_{LFP} and θ_{V_m} .

(h) Circular distribution of the θ phase difference between LFPs and V_m when θ_{LFP} and θ_{V_m} occurred simultaneously at similar frequencies (Δ frequency < 0.01 Hz). Because θ_{LFP} propagates along the mediolateral axis, the datasets were divided into two groups, in which the locations of the recorded cells were medial to the LFP recording sites (left) and vice versa (right). Red lines show the mean θ phase differences (-74° and 30° for left and right panels, respectively). The distribution was significantly nonuniform (left, $P = 8.1 \times 10^{-8}$, $Z = 16.0$, Rayleigh test, $n = 172$ periods from 30 cells; right, $P = 0.038$, $Z = 3.25$, Rayleigh test, $n = 46$ periods from 16 cells).

2- Provided LFP locations are comparable, there are some differences between

pairs of electrodes (correlation coefficient from 0.5 to near 1.0), suggesting that comparison between cells and LFP may be confounded. First, correlation analysis of LFP signals in Fig.1 is implemented for the entire recording (and hence higher correlations are favored by the slower components). Consistently, when theta periods are more strictly selected (larger SD thresholds) correlations drop (sup.fig.2). Oddly, when similar criteria applies to intracellular signals an indirect measure of correlation rises instead (differences between the intra and extra theta frequencies becomes smaller) (Fig.3g). Thus, results appear very sensitive to the definition of the analysis windows and the location of the LFP channel. The authors should find a way to standardize comparisons.

Thank you for the valuable suggestion. Regarding the time window for the correlation analyses, we calculated the correlation coefficient between simultaneously recorded θ_{LFP} powers by using time windows of 1, 2, 3, 5, 10, and 15 minutes (Supplementary Fig. 2). The ratio of time windows with significant correlations increased as the window length increased, as you noted, and no difference was observed for time windows longer than 3 minutes (Supplementary Fig. 2, All lengths, $P = 1.1 \times 10^{-4}$, $\chi^2 = 25.6$; time windows longer than 2 min, $P = 0.049$, $\chi^2 = 9.6$; time windows longer than 3 min, $P = 0.11$, $\chi^2 = 6.11$, Chi-square test, $n = 900, 450, 300, 180, 90,$ and 60 time windows for 1, 2, 3, 5, 10, and 15 min, respectively). However, even with 1-minute time windows, significant positive correlations were observed in more than 90% of the recording periods, indicating that the impact of the window length on the θ_{LFP} power correlation was not significant. We added the description to the Results section (LL. 104-114) and figure legends (LL. 976-987).

Supplementary Figure 2: Highly synchronous θ_{LFP} power changes regardless of the length of the time window.

Correlation coefficients of the θ_{LFP} power changes between all 30 pairs of simultaneously recorded LFPs were calculated by using 1-, 2-, 3-, 5-, 10-, and 15-min time windows, and the ratio of the time windows with significant correlations is shown for each window length. The ratio increased as the length of the time window increased, and no significant differences were observed for time windows longer than 3 min (all lengths of time windows, $P = 1.1 \times 10^{-4}$, $\chi^2 = 25.6$; time windows longer than 2 min, $P = 0.049$, $\chi^2 = 9.6$; time windows longer than 3 min, $P = 0.11$, $\chi^2 = 6.11$, chi-square test, $n = 900, 450, 300, 180, 90,$ and 60 time windows for 1, 2, 3, 5, 10, and 15 min, respectively). For all window lengths, the ratio exceeded 90%, indicating highly synchronous θ_{LFP} power changes regardless of the window length used to calculate the correlation coefficients.

To remove the effects of slow components from the cross-correlation analyses in Fig. 1, LFP traces were bandpass filtered before calculating the cross-correlation function (Fig. 1h, i). The correlation coefficient between pairs of θ_{LFP} powers was not affected by slow components because only the wavelet power in the θ frequency band was used.

Fig. 1: Synchronized θ_{LFP} s across the dorsal hippocampal CA1 area.

(h) Cross-correlograms of the 14 pairs of simultaneously recorded LFP traces bandpass filtered at 3-10 Hz (gray line) and their mean (black). Only the pairs of LFPs for which the relative positions of the electrodes along the mediolateral axis could be identified were included. In the inset, the time scale is expanded near 0 ms, and the plot indicates that the θ_{LFP} s recorded from relatively medial recording sites preceded their counterparts.

(i) Same as h, but for the 15 pairs of LFPs for which the relative positions of the electrodes along the anteroposterior axis could be identified. Cross-correlations peaked at 0-ms time lags.

The apparent inconsistency observed in Fig. 3g could be caused by the time windows targeted for each analysis, *i.e.*, whether the θ period was detected by using the threshold. We apologize for the misleading analyses. We reconsidered the analysis and realized that the difference in θ frequencies between the LFP and V_m should have been calculated only during the periods when the θ power of both signals were strong enough to be detected as θ periods. Therefore, we revised the analysis so that each dot indicates a “co- θ ” period during which the θ power of both signals were above the threshold; however, the results remained the same ($R = -0.094$, $P = 9.8 \times 10^{-7}$, $n = 2,702$ co- θ periods from 160 cells). The figure legend for Fig. 3g (LL. 905-908) and the related description in the Results section (LL. 199-201) were revised accordingly.

Fig. 3: Weak correlations between θ_{LFP} and θ_{V_m} .

(g) The difference in the θ frequencies of θ_{LFP} and θ_{V_m} in a co- θ period was negatively correlated with the geometric average of their powers. Each dot represents a co- θ period.

The black line indicates the line of best fit based on least-squares regression. $R = -0.094$, $P = 9.8 \times 10^{-7}$, $n = 2,702$ co- θ periods from 160 cells.

3- Results from a neural network model to predict LFP from intracellular signals should be better supported. First, why this architecture (a convolutional input layer plus deep layers)? No hyper-parameter assessment is provided. No test for overfitting. Second, how much do results depend on the criteria discussed above? The authors provide some hints in Supp.fig.7 by reporting prediction per oscillatory bands, but frequency band correlation was not shown for LFP-LFP and cell-LFP as a reference. Finally, that 3 cells provide better fit as compared to 1 or 2 cells seems relatively obvious. The control analysis in Sup.fig.8a appears a bit marginal.

We appreciate the comment and apologize for the lack of information. This comment greatly helped us in improving our paper. In our study, we first passed the input data to a convolutional layer, followed by a series of deep layers. Our model was compared with three other model architectures (Supplementary Fig. 9a). All models were trained on the data prepared for model tuning. The results showed that Model 1, which was selected as our final architecture, showed the lowest RMSE value among all channels (Supplementary Fig. 9b). In our model, the convolutional layer learns filters in the temporal dimension. Since convolutional layers are used to extract meaningful local structures, it is possible that the convolutional layers successfully extracted local oscillations in the input V_{ms} , which were important for predicting LFPs. The convolutional layer was followed by a series of deep layers. It is important to note that these deep layers have compressing and dilating architectures. By implementing this feature, the model is forced to extract and learn only the important features for predicting the LFPs. After the model architecture was selected, the model parameters were optimized based on the performance of the tuning data. As representative data, the results from four sets of parameters are shown in Supplementary Fig. 9c. We used a similar process to determine the optimal number of learning epochs based on the learning curve. The average of all traces shows that the RMSE value of the validation data hit the lowest at 5 epochs. After that, overfitting was observed as the RMSE of the validation data began to increase while the value of the training data continued to decrease. We added this description to the Results section (L. 284), Methods section (LL. 610-635) and figure legends (LL. 1092-1106).

Supplementary Figure 9: Model selection and parameter tuning for the DNN.

(a) Architectures of the four tested neural network models. The numbers indicate the dimensions of each layer. Conv: convolutional, FC: fully connected. The input layer receives three simultaneously recorded V_{ms} (V_{m1} , V_{m2} , and V_{m3}) that were bandpass filtered between 3 and 10 Hz (INPUT), and the model was trained to output the corresponding bandpass-filtered LFPs (OUTPUT).

(b) Left: Average learning curves across all 5 datasets for the four model architectures, indicated by different colors. Loss was computed as the root mean squared error (RMSE). Dotted and solid lines show training and validation loss, respectively. Right: Same as left but with an enlarged vertical axis.

(c) Left: Average learning curves across all 5 datasets for the four sets of parameters used

in the Model 1 architecture, indicated by different colors. Dotted and solid lines show training and validation loss, respectively. Right: Four sets of parameters and the validation RMSE for each set. Set 1 was selected because these parameters obtained the smallest validation loss.

As reference analyses for the prediction in frequency bands other than θ , we calculated the cross-correlation between pairs of bandpass-filtered LFP traces (as in Fig. 1h, i) and the correlation coefficients between LFP and V_m power changes for the single and triple patch-clamp datasets (as in Figs. 3d and 4e). For three frequency bands, *i.e.*, high gamma, low gamma, and slow oscillations, the correlations were similar to those of the θ frequency band (Supplementary Fig. 12a-c). However, we found that only θ power was high among other frequency bands from 0.5 to 100 Hz in both the LFP and V_m used for the DNN analyses (Supplementary Fig. 12d, e). Therefore, the predictability of the LFP traces from the V_m traces was applicable only for physiologically dominant oscillations, suggesting that the DNN can extract biologically significant signals. We added this description to the Results section (LL. 323-333) and figure legends (LL. 1131-1145).

Supplementary Figure 12: Weak power but similar temporal relationships among LFPs and V_m s in frequency bands other than θ .

(a) Correlated high gamma powers among LFPs and V_m s. Left: Cross-correlograms of the 30 pairs of simultaneously recorded LFPs bandpass filtered at 60-100 Hz (gray line) and their mean (black). Middle: Cumulative probability distribution of the correlation coefficients between θ_{LFP} and θ_{V_m} powers of all 24 cells used in the DNN analysis in Fig. 5. Right: Cumulative probability distribution of the correlation coefficients between θ_{LFP} powers and the mean θ_{V_m} powers of three simultaneously recorded cells for all 8 datasets used in the DNN analysis. Each red dot indicates a dataset with a significant positive correlation.

(b) Same as a, but for the low gamma frequency band (25-55 Hz).

(c) Same as a, but for the slow oscillations (0.5-1 Hz).

(d) LFP power spectrum averaged across all 8 datasets used in the DNN analysis. The most dominant peak was observed between 3 and 10 Hz.

(e) Same as d, but for V_m . Dominant peaks were observed between 3 and 10 Hz.

Regarding Supplementary Fig. 8a (Supplementary Fig. 13a in the revised manuscript), we apologize for the misleading explanation of the figure. In this figure, we calculated the correlation coefficient between the θ_{LFP} trace and the averaged θ_{V_m} trace of 1, 2, or 3 cells. The distribution of the correlation coefficients was plotted as a cumulative curve, with the light gray and black lines indicating the distributions for 1 and 3 cells, respectively. Therefore, the plot indicates that the average θ_{V_m} traces of 3 cells showed the lowest similarity with the θ_{LFP} traces, which might be the opposite result as described in your comment. We believe that this result is interesting because it suggests that θ_{LFP} traces could not be predicted by linearly summing the θ_{V_m} traces of multiple cells. We added more detailed descriptions to the figure legends (LL. 1150-1153).

Supplementary Figure 13: Correlations of θ traces and powers between LFPs and

Vms.

(a) Cumulative probability distributions of the correlation coefficients between θ_{LFP} traces and the mean θ_{Vm} traces of 1, 2, or 3 cells simultaneously recorded. Darker colors indicate the results for a larger number of cells. The correlation coefficients decreased as the number of cells increased, indicating that the mean θ_{Vm} traces of more cells were less similar to the θ_{LFP} trace. D_1 vs. 2 cells = 0.029, P_1 vs. 2 cells = 6.5×10^{-42} , D_1 vs. 3 cells = 0.049, P_1 vs. 3 cells = 4.1×10^{-59} , D_2 vs. 3 cells = 0.021, P_2 vs. 3 cells = 1.3×10^{-12} , two-sample Kolmogorov–Smirnov test, $n = 112,740, 112,740, \text{ and } 37,580$ 1-s segments from 8 mice for 1, 2, and 3 cells, respectively.

4- A main problem of the paper is the approach. Using just 3 cells plus a distant LFP make results too much dependent on variability (recording duration, location, cell identity, local vs global influences, ensembles, etc...). The manuscript will benefit from accepting and discussing these limitations. For example, recent data support the idea that cell-type specificity is a major factor influencing contribution to LFP signals due to biases in connectivity and the definition of local microcircuits. The authors refer to basket cell interneurons in a very general way, but PV and CCK differently connect to deep and superficial CA1 cells and so their contribution to LFP may differ (as already shown for sharp-wave ripples for instance, but also during theta). Similarly, cells can join dynamically to different ensembles and therefore the ability to predict distant and local LFP should also reflect these biases.

Thank you for this helpful comment. We acknowledge that our analyses ignored the recently well-described local circuit structure, such as the biased innervation from the upstream regions and local interneurons, which show phase-locking activity in a cell-type specific manner (Soltesz and Losonczy, 2018; Valero and Prida, 2018). We discussed possible mechanisms underlying the coordination between θ_{LFP} and θ_{VmS} (LL. 420-439) and the limitations of this study in the Discussion section (LL. 454-469).

Furthermore, as a possible approach to addressing the anatomical and physiological biases in our datasets, we examined the effects of the mean firing rate, mean V_m values, standard deviation of V_m , and neuron location along the radial (deep vs. superficial) and proximodistal axes (from CA1a to CA1c) on the correlation coefficients between θ_{LFP} and θ_{Vm} power changes, although we did not find any factor that explains the correlation between θ_{LFP} and θ_{Vm} (Supplementary Fig. 5c-g). We also compared the phase difference between θ_{LFP} and θ_{Vm} for deep and superficial cells, which could be affected by the different phase preferences of PV- and CCK-positive basket cells. However, Supplementary Fig. 5h shows that deep and superficial cells exhibited similar phase preferences, while a previous study reported bimodal θ_{LFP} phase preferences at the suprathreshold level (Navas-Olive et al. 2020). Because deep cells receive more inputs from the medial entorhinal cortex than superficial cells, this discrepancy might indicate that excitatory inputs from the medial entorhinal cortex, which were weakened under anesthesia, contribute to the shifted phase of deep cell spikes. We added this description to the Results section (LL. 179-191, 212-215), Discussion section (LL. 420-439) and figure legends (LL. 1018-1037).

Supplementary Figure 5: Relationships between θ_{LFP} - θ_{Vm} power correlations and physiological and anatomical properties of individual cells.

(c) No significant relationship was found between the correlation coefficients and the mean firing rate of each cell. $R = -0.081$, $P = 0.31$, $n = 160$ cells.

(d) No significant relationship was found between the correlation coefficients and the mean V_m of each cell. $R = -0.14$, $P = 0.076$, $n = 160$ cells.

(e) No significant relationship was found between the correlation coefficients and the standard deviation (SD) of V_m for each cell. $R = 0.068$, $P = 0.39$, $n = 160$ cells.

(f) No significant difference was found in the correlation coefficients depending on the cell locations along the proximodistal axis in the CA1 subregion. $P = 0.62$, one-way analysis of variance (ANOVA), $n = 27, 41, 45, 11, 11$ cells for CA1a, a/b, b, b/c, c, respectively.

(g) No significant difference was found in the correlation coefficients depending on the cell locations along the radial axis in the CA1 subregion. $P = 0.27$, $t_{131} = 1.1$, Student's t -test, $n = 80$ and 53 for deep and superficial, respectively.

(h) Circular distribution of the θ phase difference between LFPs and V_m when θ_{LFP} and θ_{Vm} occurred simultaneously at similar frequencies (Δ frequency < 0.01 Hz). The datasets were divided into two groups, in which the recorded cells were located in deep (left) or superficial (right) layers. Red lines indicate the mean θ phase differences (-64° and -63° for deep and superficial cells, respectively). No significant difference was found between the preferred phases of deep and superficial cells ($P > 0.1$, $K = 1.8 \times 10^3$, Kuiper test, $n = 151$ and 74 periods from 26 and 20 deep and superficial cells, respectively).

We agree that the dynamic participation of each cell in distinct ensembles should be taken into account when considering the relationships among distant LFPs and V_m s. However, we assume that our recordings under anesthesia capture only intrinsic cell assemblies, which are mainly defined by anatomical and physiological properties in the hippocampal circuit (L. 412). Further examinations in behaving animals are needed to capture the dynamic participation of individual cells to functionally distinct ensembles.

This point was added to the revised manuscript in conjunction with a discussion of the animal condition (LL. 397-402).

5- There is some asymmetry in the LFP-LFP cross-correlation function(Fig.1h). Is there any mediolateral, or rostro-caudal trend?

Thank you for your perceptive comment. As mentioned in the answer to Comment #1, we revised the cross-correlation analyses shown in Fig. 1. In Fig. 1h and i, we calculated cross-correlations of LFP trace pairs that were bandpass filtered at 3-10 Hz by using the LFPs recorded from relatively medial or anterior locations in each pair as references. The resulting time lags indicated that the θ_{LFP} s recorded at relatively medial positions significantly preceded their counterparts (Fig. 1j ML, $P = 0.0023$ vs. time 0, $t_{14} = -3.7$, Student's t -test, $n = 15$ pairs from 5 mice), while no such effect was observed for θ_{LFP} s along the anteroposterior axis (Fig. 1j AP, $P = 0.095$ vs. time 0, $t_{13} = 1.8$, Student's t -test, $n = 14$ pairs from 5 mice), indicating the mediolateral propagation of θ_{LFP} as described in Lubenov and Siapas, 2003. The reduced asymmetry in the revised plots compared to the original plot suggests that the asymmetry in the original LFP-LFP cross-correlation reflected mediolateral effects, as you noted. We added a description to the Results section (LL. 114-126), Discussion section (LL. 368-375), and figure legends (LL. 845-857).

Fig. 1: Synchronized θ_{LFP} s across the dorsal hippocampal CA1 area.

(h) Cross-correlograms of the 14 pairs of simultaneously recorded LFP traces bandpass filtered at 3-10 Hz (gray line) and their mean (black). Only the pairs of LFPs for which the relative positions of the electrodes along the mediolateral axis could be identified were included. In the inset, the time scale is expanded near 0 ms, and the plot indicates that the θ_{LFP} s recorded from relatively medial recording sites preceded their counterparts.

(i) Same as h, but for the 15 pairs of LFPs for which the relative positions of the electrodes along the anteroposterior axis could be identified. Cross-correlations peaked at 0-ms time lags.

(j) Time lags between pairs of θ_{LFP} s were calculated for the mediolateral (ML) and anteroposterior (AP) pairs shown in h and i, respectively. θ_{LFP} propagation was observed in the medial to lateral direction ($P = 0.0023$ vs. time 0, $t_{14} = -3.7$, Student's t -test, $n = 15$ pairs from 5 mice) but not along the anteroposterior axis ($P = 0.095$ vs. time 0, $t_{13} = 1.8$, Student's t -test, $n = 14$ pairs from 5 mice).

6- Fig.1e shows a long LFP recording period, while 2C,D is shorter. Recording

duration is limited for intracellular recordings, and so cell-LFP analysis may be influenced by this factor. Can the authors provide some equivalent assessment of LFP-LFP correlations?

Thank you for noting this important point. As mentioned in the answer to Comment #2, we calculated the correlation coefficient between simultaneously recorded θ_{LFP} powers by dividing the entire recording time into 1-, 2-, 3-, 5-, 10- or 15-minute segments (Supplementary Fig. 2). The results showed that the ratio of the time windows with significant correlations increased as the time window length increased, and no differences were observed for time windows longer than 3 minutes. However, even with 1-minute time windows, significant positive correlations were observed in more than 90% of the recording period. We added the description to the Results section (LL. 104-114) and the figure legends (LL, 976-987).

Supplementary Figure 2: Highly synchronous θ_{LFP} power changes regardless of the length of the time window.

Correlation coefficients of the θ_{LFP} power changes between all 30 pairs of simultaneously recorded LFPs were calculated by using 1-, 2-, 3-, 5-, 10-, and 15-min time windows, and the ratio of the time windows with significant correlations is shown for each window length. The ratio increased as the length of the time window increased, and no significant differences were observed for time windows longer than 3 min (all lengths of time windows, $P = 1.1 \times 10^{-4}$, $\chi^2 = 25.6$; time windows longer than 2 min, $P = 0.049$, $\chi^2 = 9.6$; time windows longer than 3 min, $P = 0.11$, $\chi^2 = 6.11$, chi-square test, $n = 900, 450, 300, 180, 90$, and 60 time windows for 1, 2, 3, 5, 10, and 15 min, respectively). For all window lengths, the ratio exceeded 90%, indicating highly synchronous θ_{LFP} power changes regardless of the window length used to calculate the correlation coefficients.

Furthermore, we examined the effect of the recording time on the correlation coefficient between θ_{LFP} and θ_{r_m} power changes (Supplementary Fig. 5a). No significant correlation was found between the recording time and the correlation coefficient for each data (Supplementary Fig. 5a, $P = 0.19$, $R = 0.11$, t -test for correlation coefficients, $n = 160$ cells). Therefore, the impact of the recording time may be small enough to not affect the outcome of this study. We added the description to the Results section (LL. 179-181) and the figure legends (LL. 1009-1014).

Supplementary Figure 5: Relationships between $\theta_{\text{LFP}}\text{-}\theta_{\text{Vm}}$ power correlations and physiological and anatomical properties of individual cells.

(a) Correlation coefficients between θ_{LFP} and θ_{Vm} power changes were plotted against the recording time for each dataset. No significant relationship was found between the correlation coefficients and recording times. $R = 0.11$, $P = 0.19$, t -test for correlation coefficients, $n = 160$ cells.

7- In general, discussion is not updated with recent hippocampal data on intracellular versus extracellular dynamics (e.g. results regarding different theta frequencies and phase shifts between intra and extracellular signals, the role of cell-type specificity, proximodistal shifts, etc....). There are many more references to cortical dynamics than to the specific dorsal hippocampus itself.

We appreciate the comment and apologize for the outdated discussion. We revised the discussion about possible mechanisms for the coordination and phase relationships between θ_{LFP} and θ_{Vm} s by considering the anatomical and physiological biases within the dorsal CA1 region (Igarashi et al., 2014; Soltesz and Losonczy, 2018; Valero and Prida, 2018; Navas-Olive et al., 2020) and previous intracellular studies (Soltesz and Deschenes, 1993; Ylinen et al., 1995; Valero et al., 2022) (LL. 420-439).

Reviewer #3

The manuscript by Noguchi et al., examines the relationship between the activity of hippocampal CA1 neurons and theta oscillations recorded in the local field potential. This study focuses on the atropine sensitive type 2 theta oscillations and demonstrates a complex relationship between individual neurons and LFP theta, whereby one population of neurons (approx. 40%) is highly correlated with LFP theta power, while the remaining neurons show no significant correlation. The authors go further by characterizing the phase relationship between VM and LFP theta oscillations, finding that even with as few as three simultaneously recorded neurons, a neural network model is able to accurately predict the amplitude and phase of extracellularly recorded theta LFPs. Clearly, the authors

have invested a significant amount of time and expertise in gathering and analyzing this extensive dataset. On the whole the results are solid and presented clearly throughout, with the conclusions drawn by the authors being strongly supported by the underlying data. I would be happy to support publication of the manuscript and have included some suggestions below, which I feel would further clarify and strengthen the authors conclusions.

We appreciate that you noted the value of our experimental dataset. Thanks to your comments, we were able to revise and improve our manuscript. Our individual responses are listed below:

1. My one major concern which I feel could be easily addressed by altering the analysis a little is the potential for over and undersampling of specific cells in some of the figures presented. As far as I understand the data shown in Figures 2f & g for example include all VM theta periods (100ms each, with each dot representing a single one). This would mean that with approx. 30 % of the time being a theta state, the shortest recorded cell (30s) would contribute about 300 data points, while the longest recorded cell (2000s) would add 20000 data points. Depending on the distribution of the recording times, this has the potential to greatly skew certain results to a handful of neurons at the upper end. To overcome this, the z-scored power for each cell could be binned, with only the average power and firing rate for each bin added on a cell by cell basis.

We appreciate the comment and apologize for the misleading description of the plots shown in Fig. 2f and g, which we believe already follow the suggestions. For Fig. 2f and g, the firing rate, θ_{VM} power and θ_{VM} frequency of each cell were Z-standardized on a logarithmic scale across the entire recording period. The average values of each parameter were subsequently calculated for each θ_{VM} period, which are each indicated as a dot and superimposed on a cell-by-cell basis. We revised the figure legends to specify the calculation process (LL. 874-876).

Fig. 2: Variable θ_{VM} frequencies in a CA1 pyramidal cell.

(f) The firing rates increased with increases in the θ_{VM} power. To pool data from different cells, the firing rates were Z-standardized on a logarithmic scale across the entire recording period of each cell. Each dot indicates the average value in a single θ_{VM} period, and the average values of the Z-standardized parameters are superimposed on a cell-by-cell basis. The black line indicates the line of best fit based on a generalized linear mixed model. $\beta = 0.22$, $P = 8.0 \times 10^{-9}$, $t_{1032} = 5.8$, t -test of the correlation coefficient, $n = 1,034$

θ_{Vm} periods from all 136 cells that fired at least one spike.

(g) The same as f but for the θ_{Vm} frequencies. $\beta = 0.10$, $P = 0.035$, $t_{1032} = 2.1$, $n = 1,034$ θ_{Vm} periods from 136 cells.

2. The methods section could be expanded a little. There seems to be no mention of how the awake recordings differed from the anesthetized recordings, how were the animals forced to be immobile in the awake state? This should be clarified.

We apologize for the lack of information. To obtain recordings from unanesthetized mice, mice were implanted with metal head-holding plates under short-term anesthesia with 2–3% isoflurane. After full recovery, the mice received head-fixation training on a custom-made stereotaxic fixture for 1–2 h per day. The training continued for up to 5 days until the mice learned to remain calm. We added the description to the Methods section (LL. 483-496).

3. Though the authors suggest in the discussion that the same principles may apply to type I theta as observed here, it should be noted that the increased amplitude and frequency of type I theta (which the authors show in this study, both increase the correlation strength between θ_{LFP} and θ_{VM}) may result in a larger fraction of the neuronal population being coupled to LFP theta.

Thank you for noting this important point. We agree with your opinion and have revised the discussion so that the similar and different aspects of the circuit dynamics during type 1 and type 2 θ_{LFP} s were both considered. We agree that different circuit mechanisms should be assumed to consider the coordinated θ_{LFP} and θ_{Vm} s in awake behaving animals. We have revised the Discussion section in the manuscript (LL. 393-402).

4. One interesting avenue to explore would be to try and determine what differentiates the 41% of neurons that significantly correlate with LFP theta power, from the remaining 59% that don't. The authors could compare mean firing rate, resting membrane potential or potentially anatomical locations of the two groups of neurons (such as superficial vs deep CA1 cells) to establish if there is a consistent factor that determines whether or not a cell is actively engaged with the ongoing theta oscillation.

Thank you for the constructive comment. We examined the effects of the mean firing rate, mean V_m values, standard deviation of V_m , and neuron location along the radial (deep vs. superficial) and the proximodistal axes (from CA1a to CA1c) on the correlation coefficients between θ_{LFP} and θ_{Vm} power changes (Supplementary Fig. 5c-g). However, we did not find any difference between the correlated and uncorrelated cells with LFPs. We assume that the engagement of each cell with the ongoing θ_{LFP} might not be intrinsically predetermined but rather be flexibly modifiable. We added these descriptions to the Results section (LL. 179-191), Discussion section (LL. 420-439), and figure

legends (LL. 1009-1037).

Supplementary Figure 5: Relationships between θ_{LFP} - θ_{V_m} power correlations and physiological and anatomical properties of individual cells.

(c) No significant relationship was found between the correlation coefficients and the mean firing rate of each cell. $R = -0.081$, $P = 0.31$, $n = 160$ cells.

(d) No significant relationship was found between the correlation coefficients and the mean V_m of each cell. $R = -0.14$, $P = 0.076$, $n = 160$ cells.

(e) No significant relationship was found between the correlation coefficients and the standard deviation (SD) of V_m for each cell. $R = 0.068$, $P = 0.39$, $n = 160$ cells.

(f) No significant difference was found in the correlation coefficients depending on the cell locations along the proximodistal axis in the CA1 subregion. $P = 0.62$, one-way analysis of variance (ANOVA), $n = 27, 41, 45, 11, 11$ cells for CA1a, a/b, b, b/c, c, respectively.

(g) No significant difference was found in the correlation coefficients depending on the cell locations along the radial axis in the CA1 subregion. $P = 0.27$, $t_{131} = 1.1$, Student's t -test, $n = 80$ and 53 for deep and superficial, respectively.

5. Since correlation strength between Θ_{LFP} and Θ_{V_m} is modulated by theta power and frequency, it might be prudent to rule out any changes in LFP theta. During the dual patch experiments there appears to be an increased percentage (64%), vs the 41% (single patch) and 43% (triple patch) of cells that show an increased correlation between V_m and LFP theta power. While I doubt that any systematic changes in LFP occurred specifically during the dual patch experiments, it would be better to rule out this as a potential problem.

Thank you for the kind suggestion. We calculated the percentage of cell pairs that showed positively correlated θ_{V_m} powers (the same analysis as Supplementary Fig. 7e in the revised manuscript) by using only the data included in the LFP-cell analysis (Fig. 3e).

However, the percentage of cell pairs with positive correlations did not change significantly from the original percentage (Supplementary Fig. 8a, b, $P = 0.49$, $\chi^2 = 0.47$, chi-square test, $n = 125$ (all data) and 98 (data in Fig. 3 only) cell pairs, respectively). Therefore, the high proportion of positively correlated cell pairs, compared to the proportions of significant datasets for single or triple patch-clamp experiments, should not be due to any systematic changes in LFPs occurring specifically during the dual patch-clamp experiments. We added this description to the Results section (LL. 240-245) and figure legends (LL. 1082-1090).

Supplementary Figure 8: Similar ratios of cell pairs with correlated θ_{Vm} power changes in the data shown in Fig. 3 and Supplementary Fig. 7.

(a) Cumulative probability distribution of the correlation coefficients between the θ_{Vm} powers of 98 cell pairs, which were included in the analyses for Fig. 3. Each red dot indicates a cell pair with a significant correlation.

(b) Ratios of cell pairs with significantly correlated θ_{Vm} power changes for the data used in the analyses for Supplementary Fig. 7 (left) and only the data used for the analyses in Fig. 3 (right). No significant difference was observed. $P = 0.49$, $\chi^2 = 0.47$, chi-square test, $n = 125$ (all data) and 98 (data in Fig. 3 only) cell pairs.

6. In the discussion the authors touch on the role of PV-positive basket cells in mediating the association of Θ_{LFP} and Θ_{VM} . While clearly beyond the scope of these data, I was wondering if they could at least discuss their thoughts on how including data from PV+ neurons might impacted the performance of their DNN.

Thank you for the valuable suggestion. We will include not only the θ_{Vms} of pyramidal cells but also the activity of other phase-locking cells to θ_{LFP} , such as PV-positive basket cells when the data is available. We expect that the prediction could benefit by including information about various types of θ_{LFP} -modulated cells. We discussed this possibility as a potential future direction (LL. 461-463).

REVIEWERS' COMMENTS:

Reviewer #2 (Remarks to the Author):

The authors have thoroughly revised the paper to address all my previous comments. I congratulate them for a very clear revision. The paper is improved and I found the current version suitable for publication.

Reviewer #3 (Remarks to the Author):

I thank the authors for comprehensively addressing my comments, as well as those of the other referees. I find the revised manuscript much improved and believe it will be of interest to the community.

Comments and Answers

Reviewer #2

The authors have thoroughly revised the paper to address all my previous comments. I congratulate them for a very clear revision. The paper is improved and I found the current version suitable for publication.

We are very pleased to know that our revision of the paper met your requirements. We would like to thank you very much for the valuable comments.

Reviewer #3

I thank the authors for comprehensively addressing my comments, as well as those of the other referees. I find the revised manuscript much improved and believe it will be of interest to the community.

Thank you very much for the positive evaluation for our revised manuscript. We appreciate your constructive comments, which have led to the great improvement of our work.